# Pan-urologic cancer genomic subtypes that transcend tissue of origin

Fengju Chen[1], Yiqun Zhang[1], Dominick Bossé[2], Aly-Khan A. Lalani [2], A. Ari Hakimi[3], James J. Hsieh [4], Toni K. Choueiri[2], Don L. Gibbons[5,6], Michael Ittmann[7] & Chad J. Creighton[1,8,9,10]

Urologic cancers include cancers of the bladder, kidney, prostate, and testes, with common molecular features spanning different types. Here, we show that 1954 urologic cancers can be classified into nine major genomic subtypes, on the basis of multidimensional and comprehensive molecular characterization (including DNA methylation and copy number, and RNA and protein expression). Tissue dominant effects are first removed computationally in order to define these subtypes, which reveal common processes—reflecting in part tumor microenvironmental influences—driving cellular behavior across tumor lineages. Six of the subtypes feature a mixture of represented cancer types as defined by tissue or cell of origin. Differences in patient survival and in the manifestation of specific pathways—including hypoxia, metabolism, NRF2-ARE, Hippo, and immune checkpoint—can further distinguish the subtypes. Immune checkpoint markers and molecular signatures of macrophages and T cell infiltrates are relatively high within distinct subsets of each cancer type studied. The pan-urologic cancer genomic subtypes would facilitate information sharing involving therapeutic implications between tissue-oriented domains.

[1] Dan L. Duncan Comprehensive Cancer Center Division of Biostatistics, Baylor College of Medicine, Houston, TX 77030, USA. [2] Department of Medical Oncology, Dana-Farber Cancer Institute, Boston, MA 02215, USA. [3] Department of Surgery, Urology Service, Memorial Sloan Kettering Cancer Center, New York, NY 10065, USA. [4] Molecular Oncology, Department of Medicine, Siteman Cancer Center, Washington University, St. Louis, MO 63110, USA. [5] Department of Thoracic/Head and Neck Medical Oncology, The University of Texas MD Anderson Cancer Center, Houston, TX 77030, USA. [6] Department of Molecular and Cellular Oncology, The University of Texas MD Anderson Cancer Center, Houston, TX 77030, USA. [7] Department of Pathology & Immunology, Baylor College of Medicine, Houston, TX 77030, USA. [8] Department of Bioinformatics and Computational Biology, The University of Texas MD Anderson Cancer Center, Houston, TX 77030, USA. [9] Human Genome Sequencing Center, Baylor College of Medicine, Houston, TX 77030, USA. [10] Department of Medicine, Baylor College of Medicine, Houston, TX 77030, USA. Fengju Chen and Yiqun Zhang contributed equally to this work. Correspondence and requests for materials should be addressed to C.J.C. (email: creighto@bcm.edu)

Cancers are typically classified based on the tissue site of origin, coupled with observable histologic features, with subsequent therapeutic decisions then following the histologic classification. At the molecular level, cancers associated with a given tissue type may represent a heterogeneous group of diseases. Unsupervised approaches to subtyping human tumors on the basis of molecular profiling data provide a tool in adding to our understanding of the genomic landscape of cancer. Genomic subtypes of cancer as defined by molecular analysis may represent disease subsets being driven by distinct pathways and processes. Data platforms for messenger RNA (mRNA) expression in particular have been used extensively in the molecular classification of cancers[1], with other data platforms representing other molecular features (e.g., DNA methylation, DNA copy alteration) also being used as these became widely available. Over the last few years, The Cancer Genome Atlas (TCGA) carried out a number of genomic studies each focusing on an individual cancer type[2–6], where classification or subtyping based on multiple molecular data platforms was typically performed. Pan-cancer analysis by TCGA of an initial set of 12 different cancer types (including bladder, kidney clear cell, and prostate) found cancers to segregate largely on the basis of cancer type as defined by tissue of origin, with one notable exception being the "squamous" genomic subtype that spanned multiple cancer types, including lung squamous, head and neck squamous, and a subset of bladder cancers[7].

Urologic cancers include cancers of the bladder, kidney, prostate and testes, all relatively common, with prostate cancer, for example, being the most common cancer in American men[8]. Urologic oncology is an established specialty within medical practice, as urologic cancers arise within the urinary tract of men and women and reproductive organs of men. Urologists have a key role in the diagnosis and treatment of all of these malignancies. However, in terms of treatment, it is well understood that the different cancer types as defined by tissue of origin would represent quite distinct diseases from each other on the basis of several factors, including histologic appearance, the presence of distinct driver mutations, varying clinical course, and different responses to systemic therapy. Within kidney cancers, the three major types—clear cell, papillary, and chromophobe—are quite distinct from each other at the histologic and molecular levels and are understood to represent truly different cancer types that warrant individual study[9, 10]. At the same time, common oncogenic processes are thought to underlie cancers of different types[11, 12].

TCGA genomic data sets—which include DNA, RNA, protein, and epigenetic data—provide a major opportunity to develop an integrated picture of commonalities, differences and emergent themes across tumor lineages[13]. With the recent conclusion of the data generation phase of TCGA, systematic analyses of the entire TCGA urologic cancer cohort would allow for comparisons and contrasts to be made between the different diseases represented. Many of the molecular differences that exist among urologic cancer types—including bladder, prostate, kidney clear cell, kidney chromophobe, kidney papillary, and testicular—could arise from their respective cells and tissues of origin[7, 10]. Understanding the mechanisms driving cancer subtypes is challenged by these lineage-specific molecular signals[14, 15]. Although cancer sample profiles would tend to segregate by associated tissue type, part of these tissue dominant effects can be removed computationally to elucidate common processes driving cellular behavior across tumor lineages.

In this study, we define nine pan-cancer molecular-based subtypes, which would transcend tumor lineage across the nearly 2000 urologic cancer cases profiled by TCGA. Data involving multiple molecular profiling platforms are used to both define

these subtypes, and to characterize them in terms of associated pathways. Common processes and pathways shared across multiple cancer types include hypoxia, metabolism, NRF2-ARE, Hippo, and immune checkpoint. Subtype-specific patterns as initially observed in TCGA are also observable in an external and independent expression data set of urologic cancers.

## Results

**TCGA cohort of urologic cancers**. TCGA collected a total of 1954 primary urologic cancer specimens (Supplementary Data 1), for which data were generated for at least one of the following molecular platforms: whole exome sequencing, DNA copy by SNP array, RNA-seq, microRNA-seq, DNA methylation array, and Reverse Phase Protein Array (RPPA). These specimens were divided between six TCGA-sponsored projects, each focusing on a specific cancer type: BLCA, corresponding to the study of bladder urothelial carcinoma ($n = 412$ cases); KICH, corresponding to kidney chromophobe ($n = 66$); KIRC, corresponding to kidney renal clear cell carcinoma ($n = 537$); KIRP, corresponding to kidney renal papillary cell carcinoma ($n = 291$); PRAD, corresponding to prostate adenocarcinoma ($n = 498$); and TGCT, corresponding to testicular germ cell tumors ($n = 150$).

As expected[7, 10, 14, 16], molecular analysis of urologic cancers revealed widespread differences associated with tissue-of-origin and with histology, which suggested the need for an alternative analytic approach to identify molecular patterns that would transcend tumor lineage. Analysis of a single marker, e.g., *PDCD1* mRNA (PD1, expressed in lymphocytes), across cancer types can serve to illustrate our overall approach to molecular subtyping (Fig. 1a). While absolute *PDCD1* expression on average varies considerably between cancer types (Fig. 1a, *left*), normalizing the expression within each cancer type would serve to subtract out such differences (Fig. 1a, *right*). Overall differences between cancer types would include any differences that would be specific to the cell- or tissue-of-origin or to the tissue microenvironment. Molecular features that show interesting patterns of variation within two or more cancer types can serve to identify cancer subsets that would span tumor lineage. In an unsupervised clustering analysis, based on the aggregated patterns of 2000 genes most variable across the cancer types, the urologic cancer cases were separated almost exclusively on the basis of cancer type when using mRNA data based on absolute expression (Fig. 1b), but were separated into subtypes that were not defined by cancer type when using expression data that were normalized within cancer type (Fig. 1c).

**Multiplatform analysis uncovers nine major genomic subtypes.** Using established analytical approaches[7, 10, 16], the 1954 TCGA urologic cancer cases were subtyped according to each of the data platforms for DNA methylation, DNA copy alteration, mRNA expression, miRNA expression, and protein expression, with the various subtype calls for each sample then being consolidated to define multiplatform-based molecular subtypes (Table 1). For methylation and expression data sets, values were first centered within each cancer type, in order that tumor lineage-specific markers would not drive the subtyping patterns. Each individual platform was used to define seven different subtypes of urologic cancer (Supplementary Fig. 1a and b), consistent with previous analyses that indicated on the order of three to seven molecular subtypes existed within a given cancer type[5, 10, 17], and where solutions greater than seven for any given platform showed no appreciable increase in the levels of consensus (Supplementary Fig. 1a). To provide an integrated level of assessment of urologic cancer molecular-based subtypes, subtype calls made by the

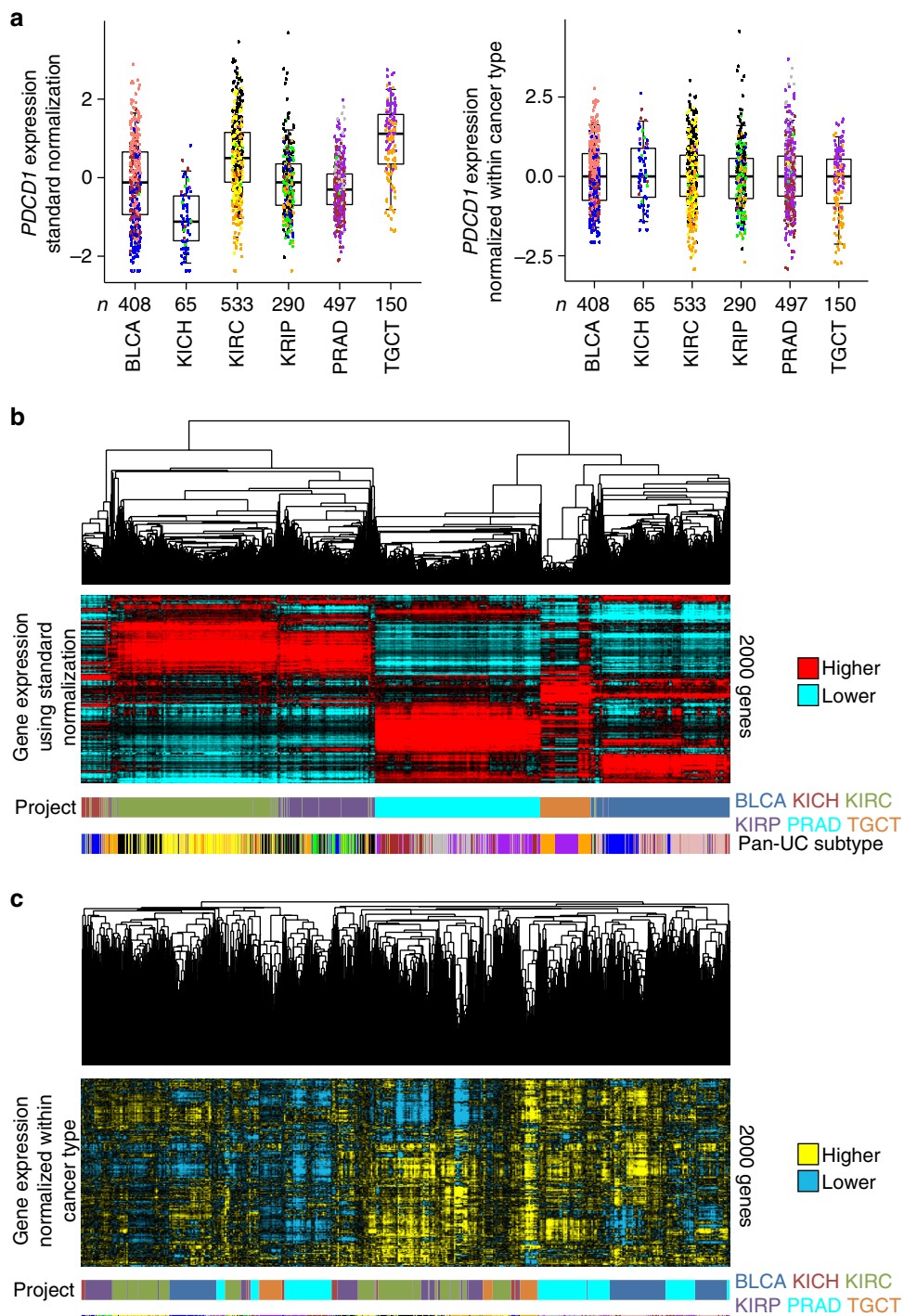

**Fig. 1** Alternative classification approach defines cancer subtypes that transcend tissue-of-origin. **a** Expression patterns of a representative marker (*PDCD1*) that distinguishes between both cancer type and genomic subtype across TCGA BLCA, KICH, KIRC, KIRP, PRAD, and TGCT cancers. *Left*, *PDCD1* expression, with values normalized using a "standard" approach (s.d. from the median across all cancers); *right*, *PDCD1* expression, with values normalized within each cancer type (giving the values within each cancer type shown a median of zero and s.d. of one). Box plots represent 5%, 25%, 50%, 75%, and 95%. **b** Unsupervised hierarchical clustering of TCGA urologic cancer cases (*n* = 1944), where expression values have standard normalization. The top 2000 most variable genes in the pan-urologic cancer data set were used in the clustering. TCGA project designation and pan-urologic cancer genomic subtype are indicated (described in Fig. 2). **c** Similar to part b, but using expression values normalized within each cancer type. The same genes from part **b** were used to carry out the clustering in part **c**. TCGA project designation: BLCA, Bladder Urothelial Carcinoma; KICH, Kidney Chromophobe; KIRC, Kidney renal clear cell carcinoma; KIRP, Kidney renal papillary cell carcinoma; PRAD, Prostate adenocarcinoma; TGCT, Testicular Germ Cell Tumors

different molecular platforms were combined by a "cluster of clusters analysis" (COCA)[7, 10, 16] approach (Fig. 2a) to form nine different integrated subtypes (Supplementary Fig. 1c–e), referred to here as "c1" through "c9".

The nine genomic subtypes of TCGA urologic cancers (Table 1) included: a c1 subtype (*n* = 216 cases) of predominantly BLCA, KICH, and KIRP cases (comprising 56%, 23%, and 19% of these cases, respectively); a c2 subtype (*n* = 333) of predominantly

**Table 1 Pan-urologic cancer genomic subtypes**

| Subtype (n cases) | Associated cancer types (%) | Associated external subtypes | Survival association | Associated pathways |
|---|---|---|---|---|
| c1 (216) | BLCA (56), KICH (23), KIRP (19) | BLCA:c1 luminal, BLCA:papillary histology, KIRP:type 1 histology | BLCA:better, KIRP:worse | fatty acid synthesis |
| c2 (333) | BLCA (28), PRAD (59), KIRP (6), KIRC (5) | PRAD:ERG fusion, PRAD:iCluster2, PRAD:meth. cluster3, BLCA:c2 lum. immune | BLCA:worse, PRAD:worse | *PTEN* loss (PRAD), Hippo |
| c3 (106) | KIRP (72), KIRC (10), PRAD (8), KICH (6) | KIRP:type 1 histology, RCC:P-e.1b | intermediate | |
| c4 (253) | PRAD (70), TGCT (28) | PRAD:ETV fusion/high, PRAD:iCluster1, PRAD:meth. cluster2, TGCT:seminoma | PRAD:worse | *SPOP* and *FOXA1* mutations (PRAD), *KIT* and *KRAS* mutations (TGCT), Hippo, immune checkpoint |
| c5 (269) | KIRC (62), KIRP (38) | RCC:CC-e.3, RCC:P-e.2, RCC:P.CIMP-e, RCC:hypermeth., KIRP:type 2 histology | KIRC:worse, KIRP:worse | fatty acid synthesis, pentose phosphate, Hypoxia, EMT, NRF2-ARE, Hippo, immune checkpoint |
| c6 (268) | KIRC (52), KIRP (18), TGCT (30) | RCC:CC-e.2, RCC:mixed, TGCT:non-seminoma | KIRC:better, KIRP:better | |
| c7 (202) | KIRC (98) | RCC:CC-e.1, RCC:CC-e.2 | intermediate | |
| c8 (192) | BLCA (99) | PanCan12:squamous, BLCA:c2 lum. immune BLCA:c3 basal BLCA:c4 immune undiff. | worse | Hypoxia, EMT, NRF2-ARE, immune checkpoint |
| c9 (115) | PRAD (100) | PRAD:ERG fusion, PRAD:iCluster3, PRAD:meth. cluster4 | better | Hypoxia, EMT, NRF2-ARE, immune checkpoint |

*BLCA* TCGA bladder project, *KICH* TCGA chromophobe renal project, *KIRC* TCGA clear cell renal project, *KIRP* TCGA papillary renal project *PRAD* TCGA prostate project, *TGCT* TCGA testicular project, *RCC*, renal cell carcinoma (KICH, KIRC, KIRP). Associated external subtypes described in Fig. 3

PRAD and BLCA cases (59% and 28%, respectively); a c3 subtype ($n = 106$) of predominantly KIRP cases (72%), with additional KIRC, PRAD, and KICH cases (10%, 8%, and 6%, respectively); a c4 subtype ($n = 250$) of predominantly PRAD and TGCT cases (70% and 28%, respectively); a c5 subtype ($n = 269$) of predominantly KIRC and KIRP cases (62% and 38%, respectively); a c6 subtype ($n = 268$) of KIRC, KIRP, and TGCT cases (52%, 18%, and 30%, respectively); a c7 subtype ($n = 202$) of predominantly KIRC cases (98%); a c8 subtype ($n = 192$) of predominantly BLCA cases (99%); and a c9 subtype ($n = 115$) of PRAD cases (100%).

The above genomic subtypes were each characterized by widespread molecular patterns. For each of the COCA-based subtypes, the top 100 genes most differentially expressed in the given subtype vs. the rest of the tumors were identified (Fig. 2b and Supplementary Data 2), where the differential patterns could be observed to span cancer type. Specific gene categories were over-represented within the top differentially expressed genes (Supplementary Fig. 2a), including extracellular- and immune-related genes being highly expressed in c5, c8, and c9 subtypes, cell cycle-related genes being highly expressed in c5 subtype, and tight junction-related genes being highly expressed in c6 and c9 subtypes. Specific proteins and

microRNAs could distinguish between the genomic subtypes (Supplementary Fig. 2b and c, respectively). Differential DNA methylation patterns could also distinguish between the subtypes (Fig. 2a and c and Supplementary Data 3), where one DNA methylation platform-specific subtype in particular spanned subtypes c1, c2, and c3. Patterns of alteration involving p16, including epigenetic silencing and copy loss—often associated with DNA hypermethylation[10]—differed by subtype (Fig. 2c). Significant anti-correlations between methylation and expression for a subset of genes could be identified (Supplementary Data 4). Within most cancer types, total DNA methylation differed according to genomic subtype (Fig. 2c and d); TGCT tumors of subtype c4 in particular showed widespread DNA hypomethylation as compared to the other cancers. Subtypes associated with high methylation included c5 (KIRC/KIRP), c8 (BLCA), and PRAD.c4 (c4 cases of PRAD type). DNA copy alteration subtypes were distinguishable on the basis of cancer type rather than genomic subtype (Fig. 2a).

**Associations with cancer type-specific subtypes.** The nine genomic subtypes made across the entire TCGA urologic cancer cohort showed high concordance with other histologic or

genomic subtype designations, which had been previously called for a subset of these cases in studies focusing on a specific cancer type[5],[10],[17] (Fig. 3a). The TCGA KIRC (kidney clear cell) cases were primarily subdivided between the c5, c6, and c7 subtypes, which largely corresponded respectively to the "CC-e.3",

"CC-e.2", and "CC-e.1" genomic subtypes previously associated with KIRC[10]. The TCGA KIRP (kidney papillary) cases were primarily subdivided between the c1, c3, c5, and c6 subtypes, with c5 associating with Type 2 histology and related "P-e.2" KIRP genomic subtype, and the other subtypes associating with Type 1

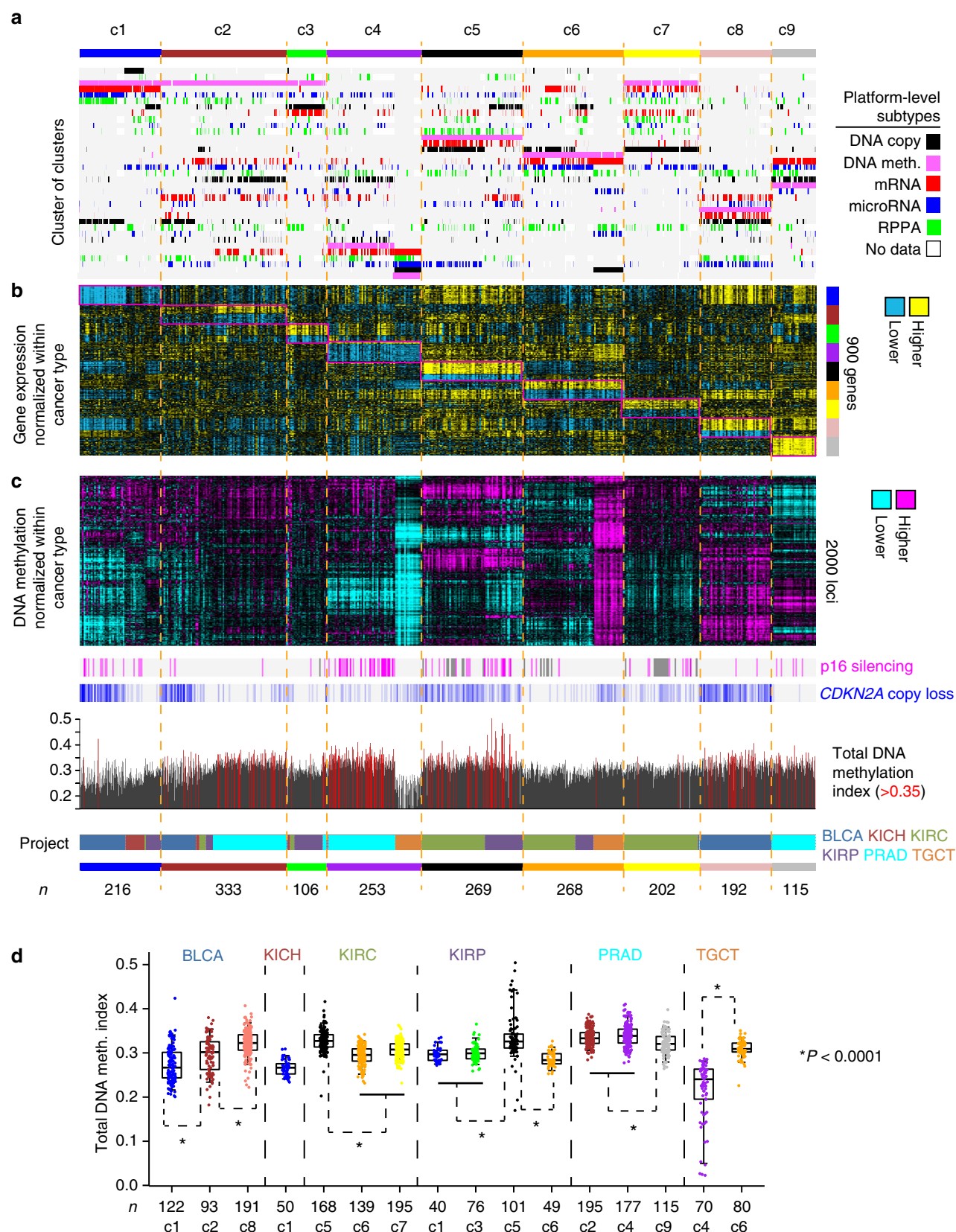

histology and related KIRP genomic subtypes. The TCGA BLCA (bladder) cases were primarily subdivided between the c1, c2, and c8 subtypes, with c1 associating with papillary histology and "luminal" expression-based subtype and with c8 associating with a "squamous" pan-cancer subtype[7]. The TCGA PRAD (prostate) cases were primarily subdivided between the c2, c4, and c9 subtypes, which largely corresponded respectively to the "iCluster2/DNA methylation3", "iCluster1/DNA methylation2", and "iCluster3/DNA methylation4" genomic subtypes previously associated with PRAD[5]. TCGA TGCT (testicular) cases were subdivided between c4 and c6, representing "seminoma" (69 out of 70 TGCT.c4 cases) and "non-seminoma" (62 out of 80 TGCT.c6 cases) histology, respectively.

For several cancer types, significant overall differences in patient survival were associated with pan-urologic genomic subtype (Fig. 3b–e). Within KIRC cases, c5, c7, and c6 were associated with worse, intermediate, and better survival, respectively (Fig. 3b, $P < 1E-10$ by Log-rank test). Within KIRP cases, c5 (associated with papillary type 2 histology) showed the worst patient survival (similar to that observed for KIRC.c5 cases), c6 showed the best patient outcome (5-year survival probability greater than 95%), and c3 and c1 showed intermediate survival (Fig. 3c, overall $P < 0.0001$). Within BLCA cases, c1 (associated with bladder papillary histology) was associated with better outcome, while c2 and c8 were associated with worse outcome (Fig. 3d, overall $P = 0.001$). Prostate cancer cases in an external gene expression profiling data set[18] (for which long-term follow-up and disease-specific survival data were available) were assigned one of the three PRAD-associated subtypes, on the basis of overall similarities in expression patterns; c9 cases showed overall better survival as compared to the other cases (Fig. 3e, overall $P < 1E-6$). In line with observations made in our previous study[10], widespread DNA hypermethylation patterns (as defined using Fig. 2c) were associated with poor patient outcomes for both KIRC and KIRP cohorts, but with no similar association being observed for BLCA cohort (Supplementary Fig. 3, there being insufficient data for evaluating the other cancer types).

Within cancer types, somatic mutation or copy alteration of specific genes could be strongly associated with pan-urologic cancer genomic subtype. For example, PRAD cases harboring *ERG* fusion or *PTEN* loss were predominantly associated with c2 subtype, and PRAD cases harboring *SPOP* or *FOXA1* mutation were associated with c4 subtype (Fig. 3a). TGCT.c4 cases were associated with *KIT* and *KRAS* mutations (Supplementary Fig. 4), while TGCT.c6 cases were associated with *KRAS* amplification. *MET* mutations were primarily associated with KIRP.c6 cases (Supplementary Fig. 4). Across the entire urologic cancer cohort, assessment of genes within pathways demonstrated a high number of alterations involving chromatin modification (41.6% of cases), p53 (34.8%), SWI/SNF complex (33.0%), Receptor

Tyrosine Kinase (RTK, 24.0%), PI3K/AKT/mTOR (21.1%), NRF2-ARE (7.3%), and Hippo signaling (3.4%) (Fig. 4a). The above pathways were found to be altered in different ways involving different genes in different cancer types (Fig. 4b and Supplementary Fig. 4). Most of the individual gene-level alterations surveyed were predominantly represented within a single-cancer type, though for some genes, e.g., *TP53* and *NF2*, mutated cases spanned multiple-cancer types (Fig. 4b).

**Distinctive biology and pathway differences across subtypes.** Analysis of gene expression data (with values normalized within cancer type) indicated differential activation of specific pathways between disease subsets. Previously, aggressive kidney clear cells cancers (associated here with our c5 pan-urologic subtype) demonstrated evidence of a metabolic shift, involving downregulation of genes involved in the TCA cycle, decreased AMPK protein levels, upregulation of the pentose phosphate pathway, and increased acetyl-CoA carboxylase protein[3]. Likewise, our c5 pan-urologic cancer genomic subtype—comprised of more aggressive cancers for both kidney clear cell and kidney papillary—showed the above patterns (Fig. 5a and Supplementary Fig. 4a, the kidney cancer cases of c6, representing less aggressive disease, being used as a comparison). TGCT of c6 subtype (associated with non-seminoma histology) also showed evidence for metabolic-related differences as compared to other TGCT, similar to those observed for c5 subtype (Fig. 5a and Supplementary Fig. 4a). Other metabolic pathway-related differences involved differences between c8 and c1—both enriched for BLCA cases, with c1 showing evidence for higher levels of fatty acid synthesis pathway (Fig. 5a and Supplementary Fig. 5a).

Clear cell kidney cancer is closely associated with *VHL* gene mutations that lead to stabilization of hypoxia-inducible factors such as HIF-1α[3]. We examined whether hypoxia-associated expression patterns may be relevant across other cancer subsets in addition to clear cell kidney. Analysis of relevant gene transcription signatures[19–22] suggested higher activation of hypoxia, epithelial-mesenchymal transition (EMT), MAP Kinase, NRF2-ARE, and cyclin D1 within the c5, c8, and c9 subtypes in particular (Fig. 5b and Supplementary Figure 5b). The coordinate manifestation of the above pathways and processes suggests previously identified inter-relationships, e.g., that of hypoxia regulating EMT[23, 24], MAP Kinase pathway[25], and NRF2-ARE pathway[26]. As Hippo pathway was previously associated with aggressive kidney papillary cancer[10], we examined transcriptional targets of Yap1, and found these to show elevated expression across c2, c4, and c5 pan-urologic genomic subtypes (Fig. 5c and Supplementary Fig. 5c).

**Fig. 2** Genomic subtypes of urologic cancers in TCGA cohort by analysis of multiple data platforms. **a** Integration of subtype classifications from five "omic" data platforms identified nine major groups of urologic cancers being represented in TCGA ($n = 1954$ cases, including bladder, renal clear cell, renal chromophobe, renal papillary, prostate, and testicular cancers). The heat map displays the subtypes defined independently by DNA methylation (*pink*), Chromosomal copy alteration (*black*), mRNA expression (*red*), microRNA expression (*blue*), and protein (RPPA) expression (*green*); each row in this heat map denotes membership within a specific subtype defined by the indicated platform. Data points are colored according to pan-urologic subtype (Table 1). **b** Differential gene expression patterns (values normalized within each main cancer type), representing a set of genes that help to distinguish between the nine subtypes (for each subtype, showing the top 100 genes most differentially in the given subtype vs. the rest of the tumors). **c** DNA methylation patterns, with heat map representing the top 2000 genomic loci with the highest variability in DNA methylation patterns across tumors (using a data set with methylation values being normalized within each main cancer type), and with corresponding p16 silencing and CDKN2A copy loss indicated, along with a plot of total DNA methylation index values (fraction of methylation probes with beta>0.3) for each sample. **d** By TCGA project and genomic subtype (where projects were highly represented within a given subtype), total DNA methylation index values. Box plots represent 5%, 25%, 50%, 75%, and 95%. *P*-values by Mann-Whitney U-test. See also Supplementary Fig. 2 and Supplementary Data 1 through 4

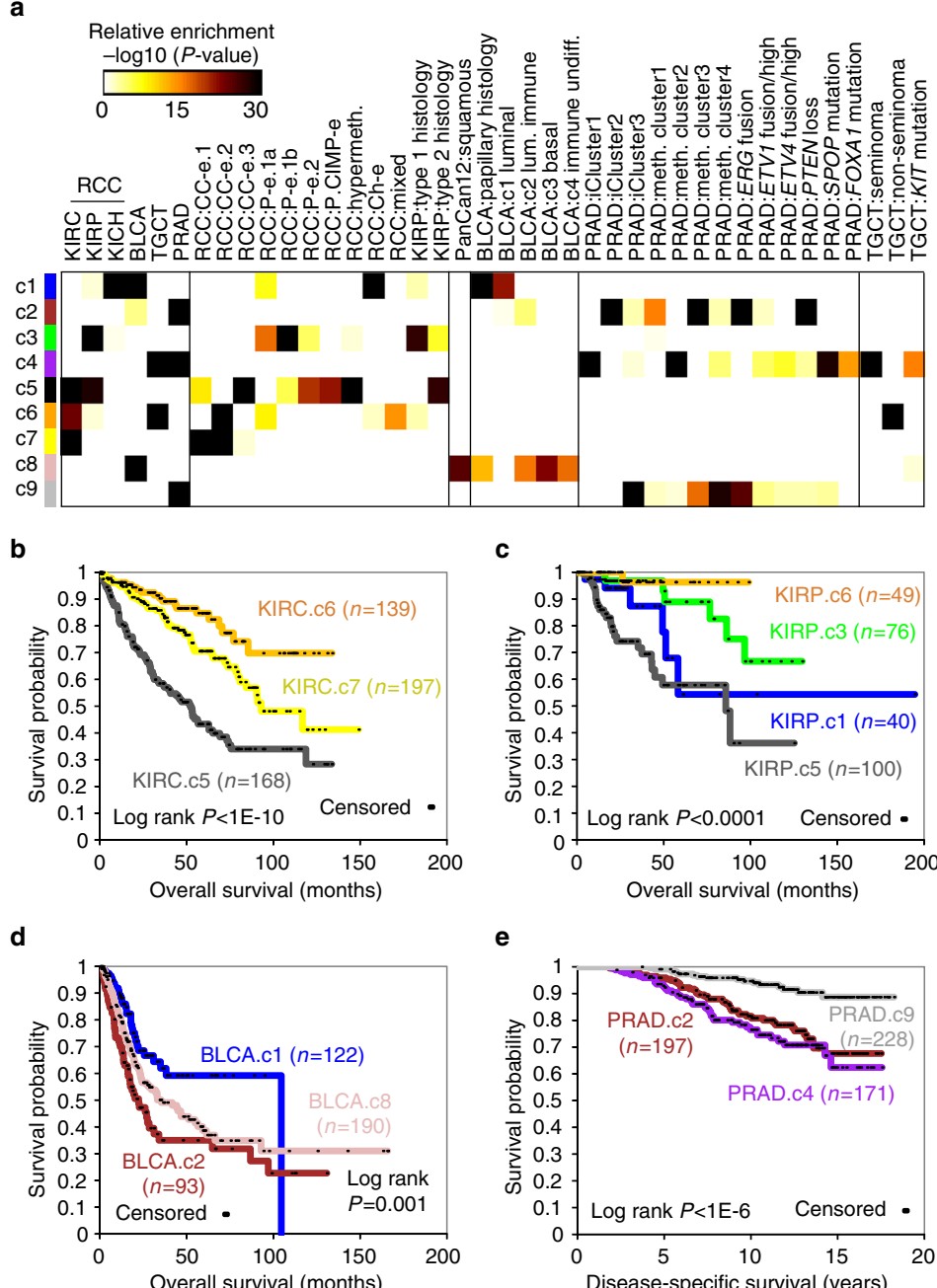

**Fig. 3** Pan-urologic cancer genomic subtypes have analogs of previously described cancer type-specific histological or molecular subtypes. **a** Significance of overlap between the pan-urologic cancer subtype assignments made in the present study (rows), with histological- or molecular-based subtype assignments (columns, also compiled in Supplementary Data 1) made previously [5, 6, 10] for a subset of cases. P-values by one-sided Fisher's exact test. RCC, renal cell carcinoma (KICH, KIRC, KIRP). "meth." and "hypermeth.", DNA methylation and DNA hyper-methylation, respectively. "lum. immune" and "immune undiff.", BLCA mRNA-based subtypes luminal immune and immune undifferentiated, respectively[17]. PanCan12:squamous associated with "squamous" pan-cancer subtype by Hoadley et al.[7] **b** Within TCGA KIRC cases, differences in patient overall survival among the pan-urologic c5, c6, and c7 genomic subtypes. **c** Within TCGA KIRP cases, differences in patient overall survival among the pan-urologic c1, c3, c5, and c6 genomic subtypes. **d** Within TCGA BLCA cases, differences in patient overall survival among the pan-urologic c1, c2, and c8 genomic subtypes. **e** Within an independent cohort of prostate cancer cases[18], classified according to our pan-urologic genomic subtypes, differences in patient overall survival among the c2, c4, and c9 subtypes. For parts b-e, P-values by log-rank test. See also Supplementary Fig. 3

**Differential immune profiles across pan-urologic subtypes.** Analysis of gene expression data (with values normalized within cancer type) indicated the presence of tumor-associated macrophages and of immune response pathways within disease subsets. In one approach, we examined the top 900 differential mRNAs (from Fig. 2b) in normal tissues, using a public expression data set from the Fantom consortium of 889 profiles representing various human cell and tissue specimens[27]. Inter-correlations between Fantom profiles and TCGA profiles suggested the presence of immune response-related cells and macrophages—along with mesenchymal-associated patterns—within genomic subtypes c4, c5, c8, and c9 (Fig. 6a). In another

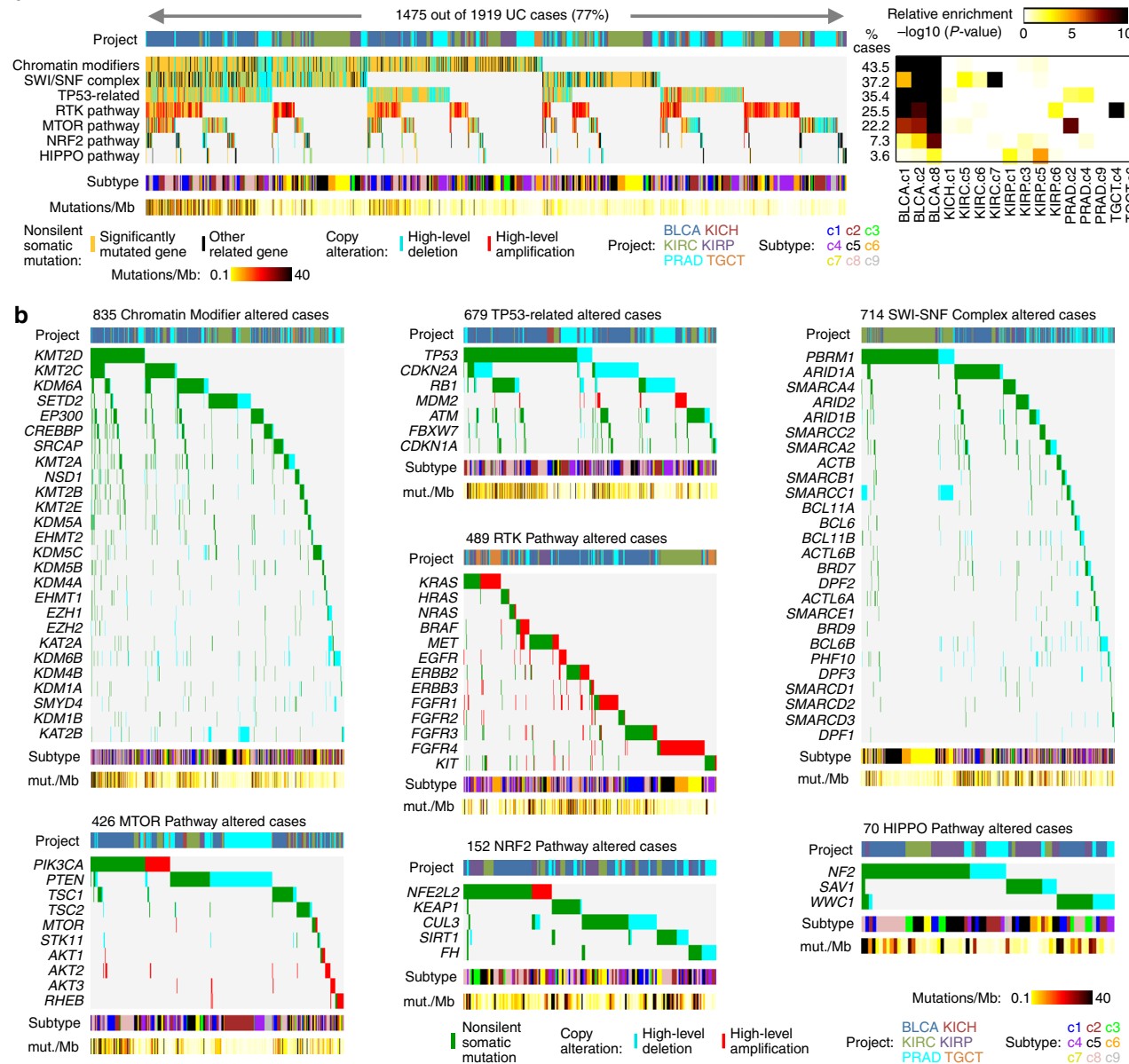

**Fig. 4** Somatic mutations and associated pathways across pan-urologic cancer types and genomic subtypes. **a** Pathway-centric view of nonsilent gene mutations and copy alterations in urologic cancers (n = 1919 urologic cancer cases with available exome sequencing data), involving key pathways and significantly mutated genes previously implicated in these cancers [5, 6, 10, 52]. Panel on the *right* represents significance of enrichment (one-sided Fisher's exact test) of gene alteration events for each pathway within any particular genomic subtype/cancer type vs. the rest of the tumors. **b** For the pathways represented in part a, mutation and copy alteration events involving each gene included in the pathway. For parts **a** and **b**, "high-level" deletion and "high-level" amplification respectively approximate total copy loss and copy levels more than 2× greater than that of wild-type (based on GISTIC thresholded values). See also Supplementary Fig. 4 and Supplementary Data 5

approach, we went on to survey the TCGA urologic cancer cases for expression of genes involved in immune checkpoint pathways (Fig. 6b). Pan-urologic c5, c8, and c9 subtypes all had relatively higher expression of several genes representing potential targets for immunotherapy[10, 28–33], including *PDCD1* (PD1), *CD247* (CD3), *CD274* (PDL1), *PDCD1LG2* (PDL2), *CTLA4* (CD152), *TNFRSF9* (CD137), *TNFRSF4* (CD134), and *TLR9*. Analysis of gene expression signatures from Bindea et al.[34] suggested that levels of immune cell infiltrates were also highest in c5, c8, and c9 subtypes (Fig. 6b and Supplementary Fig. 6). TGCT.c4 cases also showed high expression of immunotherapy target genes and T-cell infiltration, consistent with lymphocytic infiltration being associated with seminoma tumors, while PRAD.c4 cases showed

levels of immune response that were intermediate between those of PRAD.c9 and of PRAD.c2 cases.

The above associations were consistent with a consensus model of tumor-associated macrophage (TAM) roles in the tumor microenvironment[35], involving specific genes differentially expressed within c5, c8, or c9 subtypes in particular (Fig. 6c). In this model, monocytes are recruited to the tumor microenvironment by, for example, *CSF1* and *CCL2*. Cytokines secreted by the tumor then have the potential to polarize recruited monocytes into TAMs, which play vital roles in tissue remodeling, invasion and metastasis, immune suppression, and EMT. Immune suppression may involve the immune checkpoint pathway (Fig. 6d), whereby c5, c8, and c9 subtypes tend to show

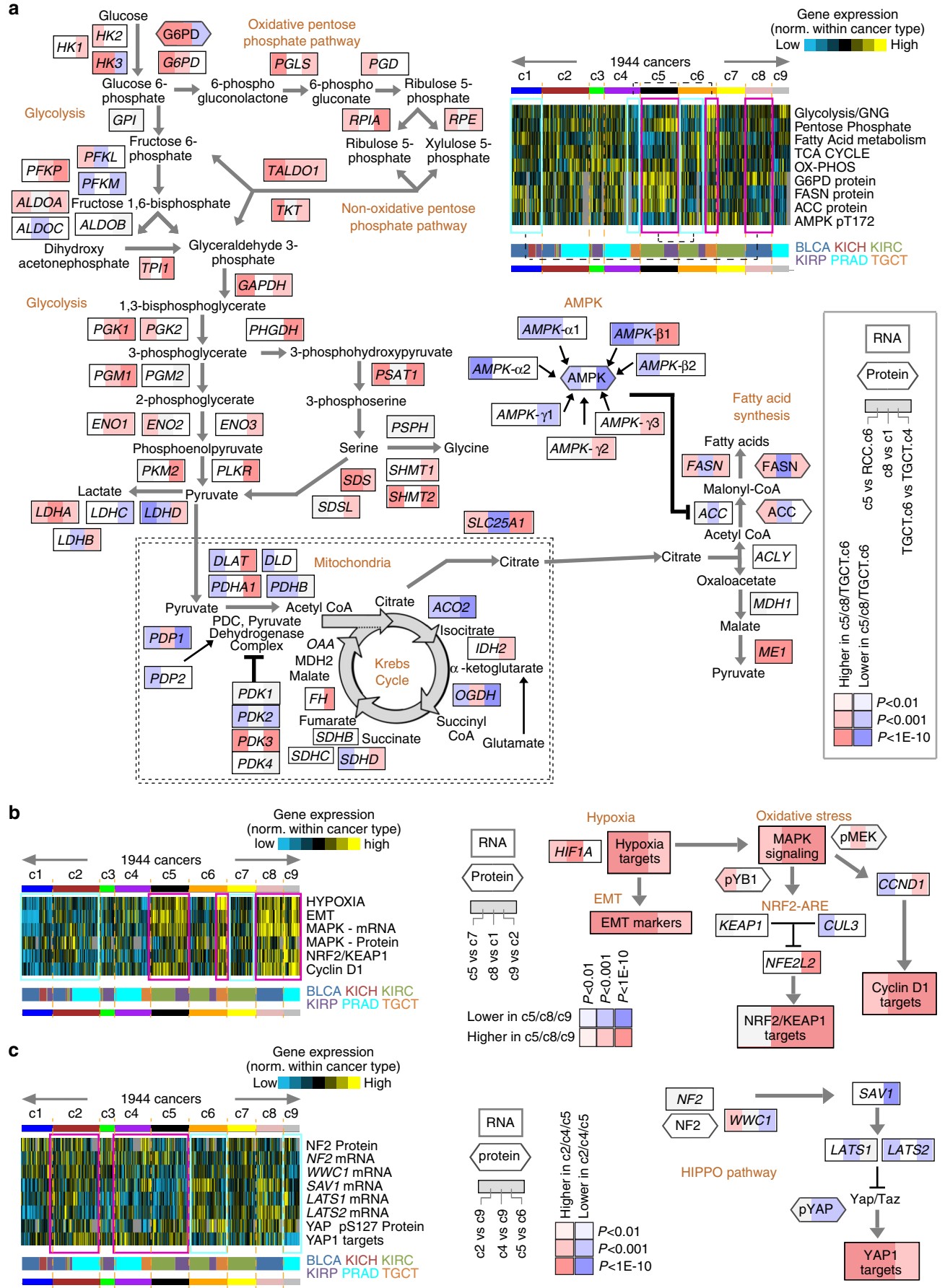

high expression of ligands such as PDL1 and PDL2 (presumably expressed by cancer cells), as well as high expression of the corresponding receptors associated with T cells.

**TCGA patterns observable in an external expression data set**. The pan-urologic genomic subtype associations, as observed in TCGA data sets, were also examined in an external multi-cancer expression data set. From the Expression Project for Oncology (expO) data set (Gene Expression Omnibus accession number GSE2109), mRNA expression profiles of bladder cancer ($n = 31$ cases), prostate cancer ($n = 83$), clear cell renal cell carcinoma ($n = 205$), and papillary renal cell carcinoma ($n = 22$) were obtained. Within each cancer type, genes in the expO data set were first normalized, and then each expO tumor profile was classified by pan-urologic genomic subtype as defined by TCGA data (Fig. 7a), where the top set of 900 mRNAs distinguishing between our nine COCA subtypes (Fig. 2b) was used as the classifier (see Methods). In the same manner as carried out for TCGA data sets (Figs. 5 and 6), expO expression profiles were also scored for mRNA signatures related to immune cells, metabolism, hypoxia, EMT, NRF2-ARE, and Yap1 (Fig. 7b), where similar overall trends of pathway-level differences between subtypes as originally observed in TCGA cohort could also be observed in the expO cohort.

**Discussion**

Our results provide a framework by which we can relate disease subsets of different urologic cancer types to each other. We identified nine major genomic subtypes of urologic cancer present in TCGA cohort (Table 1), including six subtypes which featured a mixture of represented cancer types as defined by tissue or cell of origin. Bladder cancer, clear cell kidney cancer, and prostate cancer cases were each primarily subdivided among three different subtypes (with bladder cases of papillary histology separating out from other bladder cases); papillary kidney cancer cases were subdivided among four subtypes (involving differences between type 1 and type 2 histology, for example); and testicular cancer cases were divided into two subtypes (on the basis of seminoma vs. non-seminoma histology). Molecular differences related to a number of pathways or processes—including metabolism, hypoxia, EMT, NRF2-ARE, Hippo, tumor-associated macrophages, and immune suppression—could further distinguish the subtypes, with, for example, a genomic subtype consisting of the most aggressive subsets of both clear cell and papillary kidney cancers showing deregulation for all of the above.

Molecular subtypes previously identified within each of the individual tissue-based cancer types examined here are also found within our pan-urologic cancer genomic subtypes, with, for example, the three groups of kidney clear cell cases identified in the present study corresponding to the three subtypes of more aggressive, less aggressive, and intermediate diseases identified

previously[3, 10]. This would offer opportunities to apply what has been previously learned about one subtype to another subtype, where both subtypes may originate from different tissues or cells of origin but yet associate with each other as part of a pan-urologic subtype. By this approach, for example, the aggressive subset of kidney papillary cancer was found here to have similar pathway-level associations as those of aggressive kidney clear cell cancer. For individual cancer types, specific pathways may be most relevant and would not necessarily span multiple tumor lineages, e.g., androgen signaling in prostate cancer. At the same time, given the various analytical methods and data platforms that might be used to classify cancers, the strong overlaps between previous classifications and those made by our study (Fig. 2a) would serve to better define the precise number of biologically distinct, global molecular subtypes that may be found within each cancer type.

Commonalities observed within a number of our pan-urologic subtypes would appear to have more to do with microenvironmental influences (e.g., cancer-associated macrophages and lymphocytes, altered metabolism, hypoxia, cellular stress), than with somatic mutation patterns. While the six major cancer types examined here share the same anatomic region, the tissues and cells composing each organ would be quite different. The different tissue-based cancer types would not necessarily share common etiologic factors or mutational signatures[36] (e.g., such as lung and bladder cancers both involving cigarette smoking), nor would a common "driver" mutation be found within the different cancer types comprising a pan-urologic subtype. Somatic DNA alterations may serve to help drive microenvironmental-associated phenomenon, e.g., *VHL* mutations in clear cell kidney promoting expression of hypoxia-inducible genes. However, gene mutation information alone may be insufficient to predict pathway-level alterations and microenvironmental influences as identified through integrated molecular analysis. Across the c4, c5, c8, and c9 pan-urologic subtype in particular, global differences suggested the involvement of macrophages and lymphocytes. Current understanding implicates macrophages in complex and diverse tumor-promoting roles within the tumor microenvironment, and tumor-associated macrophages would represent a potential therapeutic target[35]. Our results would suggest an intriguing hypothesis, whereby specific subtypes of urologic cancers (including subsets of kidney, bladder, testicular, and prostate cancers) would be most responsive to such a therapeutic approach.

The molecular differences represented by our pan-urologic genomic subtypes would point to pathways spanning tumor lineage and having implications for targeted therapy, including NRF2-ARE[37], Hippo[38], metabolism[39], and immune checkpoint[40, 41]. Reprogramming of energy metabolism and evading immune destruction in particular have gained attention in recent years as general hallmarks of cancer[12]. Interestingly, when diverse cancer types are analyzed in terms of absolute levels

**Fig. 5** Differentially active pathways across pan-urologic genomic subtypes. **a** Across the genomic subtypes, heat map represents differential aggregate expression of mRNAs associated with metabolism pathways, along with expression of metabolism-related proteins (using values normalized within cancer type; *gray*, no data; *GNG*, gluconeogenesis). Pathway diagram represents core metabolic pathways, with differential expression patterns comparing tumors in groups c5, c8, or TGCT.c9 (subset c9 of TGCT type) with tumors in groups RCC.c6 (c6 of KIRC or KICH type), c1, or TGCT.c4, respectively (*red*, significantly higher in c5/c8/TGCT.c9). **b** Across the genomic subtypes, heat map represents differential aggregate expression of mRNAs associated with hypoxia, EMT, MAPK, NRF2-ARE, and cyclin D1 pathways, along with a proteomic MAPK signature (using values normalized within cancer type). Pathway diagrams represent relationships between the above pathways[23–26, 53, 54], with differential expression patterns comparing tumors in groups c5, c8, or c9 with tumors in groups c7, c1, or c2, respectively (*red*, significantly higher in c5/c8/c9). **c** Across the genomic subtypes, heat map represents differential mRNA and protein features involved with Hippo pathway. Pathway diagrams represent differential expression patterns comparing tumors in groups c2, c4, or c5 with tumors in groups c9, c9, or c6, respectively (*red*, significantly higher in c2/c4/c5). See also Supplementary Fig. 5

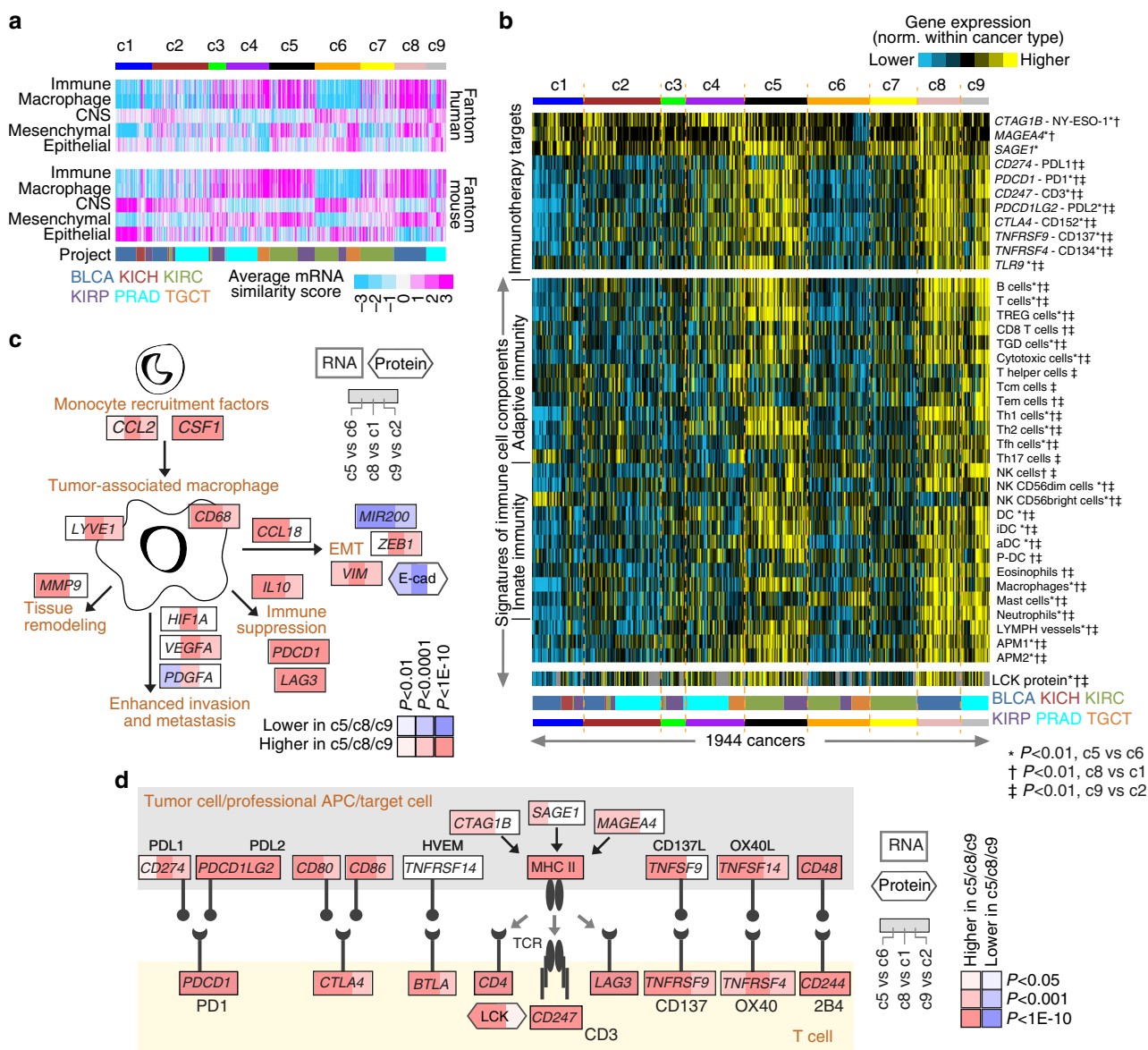

**Fig. 6** Macrophage- and immune checkpoint-related differences across pan-urologic genomic subtypes. **a** Average expression similarity correlation (Pearson's *t*-statistic, based on genes in Fig. 1b; *pink*, positive or similar; *blue*, negative or dissimilar) between TCGA urologic cancer subtypes (columns) and fantom[27] cell types or tissues in selected categories (rows): "immune," immune cell types or blood or related tissues; "CNS," related to central nervous system including brain; "macrophage," "mesenchymal," "epithelial," denoting fantom profiles for samples including one of these keywords. Results shown for both fantom human and fantom mouse data sets. For "immune," "macrophage", and "mesenchymal" correlations, differences are significant with P < 1E-10 (*t*-test), when comparing c5 with c6 cases, comparing c8 with c1 cases, and comparing c9 with c2 cases. **b** Heat maps of differential expression, for genes encoding immunotherapeutic targets (*top panels*) and for gene expression-based signatures[34] of immune cell infiltrates (*bottom panels*), across TCGA urologic cancer subtypes (expression values normalized within cancer type). *TREG cells* regulatory T cells, *TGD cells* T gamma delta cells, *Tcm cells* T central memory cells, *Tem cells* T effector memory cells, *Tfh cells* T follicular helper cells, *NK cells* natural killer cells, *DC* dendritic cells, *iDC* immature DCs, *aDC* activated DCs, *P-DC* plasmacytoid DCs, *APM1/APM2* antigen presentation on MHC class I/class II, respectively. **c** Diagram of tumor-associated macrophage roles in the tumor microenvironment[35], with differential expression patterns represented, comparing tumors in groups c5, c8, or c9 with tumors in groups c6, c1, or c2, respectively (*red*, significantly higher in c5/c8/c9). P-values by *t*-test. **d** Diagram of immune checkpoint pathway (featuring interactions between T cells and antigen-presenting cells, including tumor cells), with differential expression patterns represented, comparing tumors in groups c5, c8, or c9 with tumors in groups c6, c1, or c2, respectively (*red*, significantly higher in c5/c8/c9). P-values by *t*-test. See also Supplementary Fig. 6

of gene expression, kidney clear cell cancers as a group score among the highest in terms of immune cell infiltrates and checkpoint pathway expression[10, 16, 40]. However, within several other cancer types—including prostate, bladder, and kidney papillary cancers—there is clearly a range in scores observable across a set of cases[40]. In our study, the changes underlying the immune infiltrates and related pathways involve coordinate expression of large numbers of genes, indicative of the associated processes being systematically at work within distinct subsets of each cancer type analyzed. At what levels would the relative differential patterns observed denote real therapeutic responses in the clinical setting remains to be determined, though previous

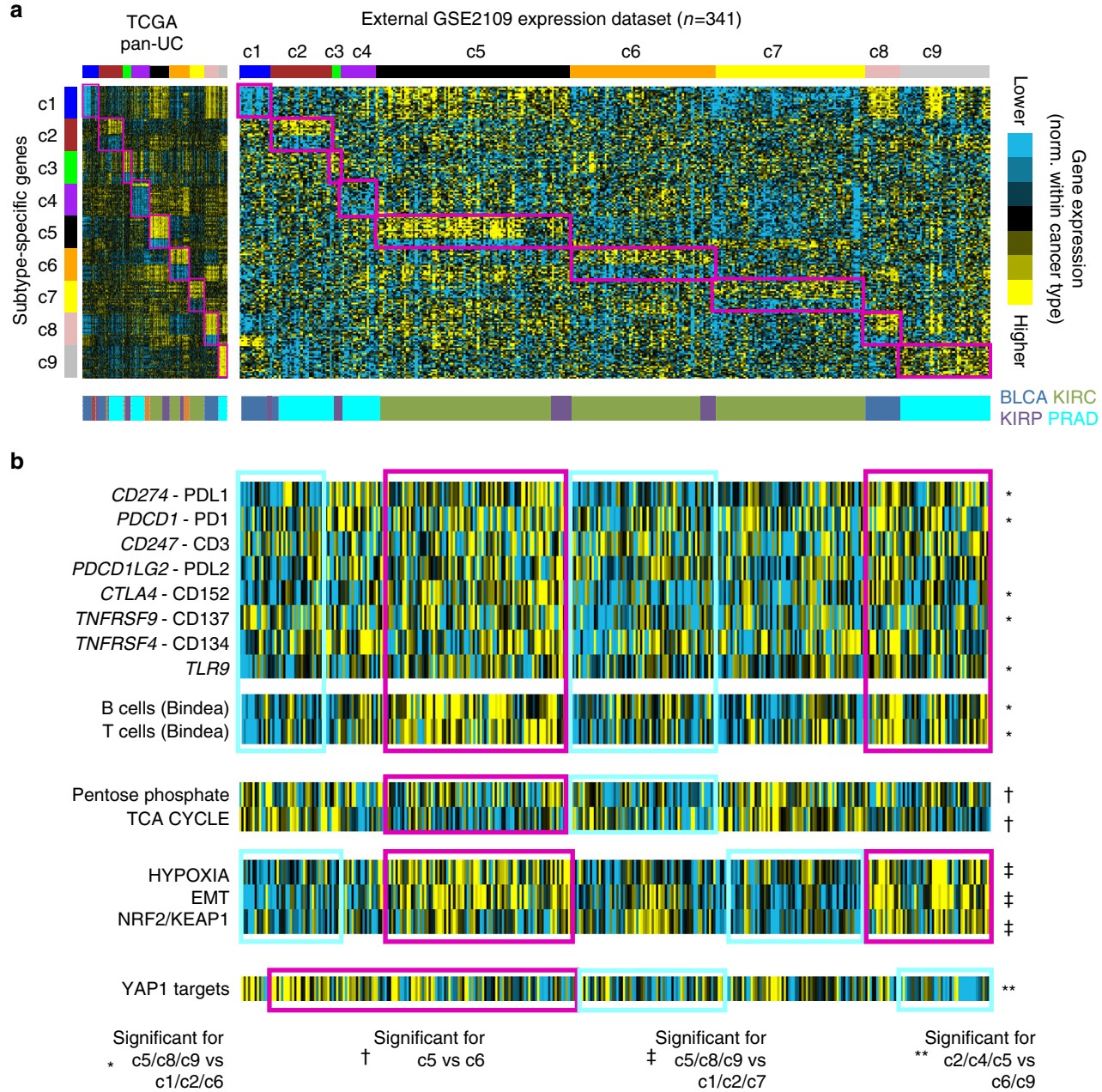

**Fig. 7** Observation of patterns associated with TCGA pan-urologic genomic subtypes in an external multi-cancer expression profiling data set. **a** Gene expression profiles of 341 urologic cancer cases (bladder, prostate, clear cell renal, papillary renal), represented in the Expression Project for Oncology (expO) (GSE2109) data set, were classified according to TCGA pan-urologic genomic subtype. Expression patterns for the top set of 900 mRNAs distinguishing between the nine COCA-based TCGA genomic subtypes (from Fig. 2b) are shown for both TCGA and GSE2109 data sets. Genes in the GSE2109 sample profiles sharing similarity with TCGA pan-urologic subtype-specific signature pattern are highlighted. **b** In the same manner as carried out for TCGA data sets (Figs. 5 and 6), expO expression profiles were scored for gene signatures related to immune cells, metabolism, hypoxia, EMT, NRF2-ARE, and Yap1. As indicated, sample profiles highlighted in *red* were compared with the sample profiles highlighted in *blue* (the comparisons being based on those carried on in Figs. 5 and 6 for TCGA data). *P*-values by *t*-test, with *P* < 0.05 considered as statistically significant. Parts **a** and **b** have the same ordering of expO expression profiles

studies indicate that immunogenicity of tumors cannot be explained by mutation load or neo-antigen load, and that expression-based markers should also be considered in the clinical setting[16, 40, 42]. The metabolic shift identified here as being associated with poor prognosis kidney cancers represents a Warburg-like metabolic phenotype (increased glycolysis, decreased AMPK, glutamine-dependent lipogenesis)[3]; such aspects of cancer cell metabolism are critical for tumor maintenance in this disease subset, yet are less likely to be relevant in

normal cells, representing yet another potential therapeutic target[39].

## Methods

**TCGA data sets**. Multiplatform genomics data sets were generated by TCGA Research Network (http://cancergenome.nih.gov/). Cancer molecular profiling data were generated through informed consent as part of previously published studies[2–6] and analyzed in accordance with each original study's data use guidelines and restrictions. In total, 1954 urologic cancer cases assayed on at least one molecular profiling platform (RNA sequencing, DNA methylation arrays, miRNA

sequencing, Affymetrix SNP arrays, whole exome sequencing, Reverse Phase Protein Arrays) were included in the analysis. Somatic mutation calls were obtained from the publicly available "MC3" TCGA MAF file (https://www.synapse.org/#!Synapse:syn7214402). All other molecular, clinical and pathological data are available through the TCGA Data Commons (https://gdc.nci.nih.gov/). For purposes of unsupervised clustering and downstream correlative analyses, with the three expression platforms (mRNA, miRNA, protein), values for each gene were centered to s.d. from the median within each cancer type (BLCA, KICH, KIRC, KIRP, PRAD, TGCT); for DNA methylation platform, beta values were centered to the median within each cancer type.

**SNP array-based copy number analysis.** DNA from each tumor or germline-derived sample had been previously hybridized by TCGA to Affymetrix SNP 6.0 arrays[3, 43]. Significant focal copy number alterations were identified from segmented data using GISTIC 2.0.22. The Broad Institute's Firehose pipeline (http://gdac.broadinstitute.org/) first filtered out normal samples from the segmented copy-number data by inspecting the TCGA barcodes and then executed GISTIC (Firehose task version: 140). We calculated copy number-based clusters for the combined pan-urologic cancer cohorts, using ConsensusClusterPlus R-package[44] to identify clusters in the data using 1000 iterations, 80% sample resampling from 2 to 15 clusters (k2 to k15) using hierarchical clustering with Ward as the linkage algorithm and Pearson correlation as the similarity metric, with an input log2 tumor:normal data set (from Firehose's "copy by gene" results table, collapsing values into cytobands). Consistent with what was carried out for the other platforms, the $k = 7$ clustering solution was selected for further investigation. High-level copy gain or copy loss events for individual genes were inferred using Firehose's "thresholded by genes" results table (+2 values being indicative of gains greater than 1–2 copies, −2 values being indicative of near total copy loss).

**Whole exome analysis.** Somatic mutation calls were obtained from the publicly-available "MC3" TCGA MAF file (representing 1769 of the 1954 patients included in this study, https://www.synapse.org/#!Synapse:syn7214402). This MC3 set is a re-calling of uniform files from all TCGA projects, with variant calling using a standardized set of mutation callers. The BAM files used underwent a standardized local re-alignment to hg19 (Genome Reference Consortium GRCh37), six calling algorithms were applied, and a number of automated filters were applied. Variants called by two or more algorithms were used in the study. For an additional 150 cases not represented in MC3 data set but with exome data featured in Chen et al.[10], variants calls from the Chen study were included in the present study. Silent mutations (point mutations that would not result in a change in the amino acid) were not included in the results. Except for *FGFR*, *KIT*, and *MET* genes, mutations in oncogenes (e.g., *KRAS*) were represented in figures, if the mutations occurred in "hotspot" residues as reported by Chang et al.[45]; all nonsilent mutations in putative tumor suppressor genes (e.g., *TP53*) were represented in figures.

**Array-based DNA methylation assay.** DNA methylation profiles had been previously generated by TCGA using either the Illumina Infinium Human-Methylation450 (HM450) or HumanMethylation27 (HM27) BeadChips (Illumina, San Diego, CA)[3] ($n = 1952$ urologic cases having methylation data). To correct for batch effects between data platforms (HM450 vs. HM27), we used the combat software[46].

For the unsupervised clustering analysis, we removed from consideration DNA methylation probes located on X or Y chromosomes and probes not included on the HM27 platform. For unsupervised clustering, we selected the top 2000 probes having the higher average variability (by s.d.) across the six projects (for projects with both HM450 and HM27 platforms, the centroid mean of the s.d. computed for each of the two platforms was used), and centered the methylation beta values to the median within each cancer type (BLCA, KICH, KIRC, KIRP, PRAD, TGCT). ConsensusClusterPlus R-package[44] was used to identify clusters in the data using 1000 iterations, 80% sample resampling from 2 to 15 clusters (k2 to k15) using hierarchical clustering with Ward as the linkage algorithm and Pearson correlation as the similarity metric. Consistent with what was carried out for the other platforms, the $k = 7$ clustering solution was selected for further investigation.

The DNA methylation level as interrogated by cg13601799 was used for *CDKN2A*; as done previously[2–4, 10], a beta value of 0.2 or above was considered evidence for epigenetic silencing. For each sample, a total DNA methylation index value was computed as the fraction of methylation probes (out of all HM27 probes) having beta value>0.3.

**Analysis of tumor features by mRNA platform.** RNA sequencing data had been previously generated by TCGA[3]. Expression of coding genes was quantified for 20,531 features based on the gene models defined in the TCGA Gene Annotation File (GAF) (https://tcga-data.nci.nih.gov/tcgafiles/ftp_auth/distro_ftpusers/anonymous/other/GAF/GAF_bundle/outputs/). Expression values within each cancer type (BLCA, KICH, KIRC, KIRP, PRAD, TGCT) were normalized to s.d. from the median across tumors; other than the analysis in main Fig. 1b, all downstream analyses using the mRNA expression data were based on the data set

with values normalized within each cancer type. For unsupervised clustering analysis, we selected the top 2000 most variable genes, according to average s.d. (using log-transformed expression values) across the six projects. Consensus ward linkage hierarchical clustering identified $k = 2$ to $k = 15$ subtypes, with the stability of the clustering increasing with increasing $k$. Consistent with what was carried out for the other platforms, the $k = 7$ clustering solution was selected for further investigation.

We examined an external gene expression profiling data set of prostate cancer from Nakagawa et al. (GSE10645)[18], classifying each external tumor profile by genomic subtype as defined by TCGA data. As a classifier, the top set of 900 mRNAs distinguishing between the nine COCA subtypes in TCGA was used. For the three COCA subtypes with high representation of PRAD cases (namely, c2, c4, and c9), the average centroid for each gene was computed, based on the centered TCGA expression data matrix (represented in Fig. 2b). Expression values for the GSE10645 data set, corresponding to the above 900 genes were centered to s.d. from the median across sample profiles. The correlation between each GSE10645 profile and each TCGA subtype centroid was computed, with the centroid showing the highest correlation (c2, c4, or c9) used to assign a TCGA-based subtype to the GSE10645 profile.

**MicroRNA analysis.** MiRNA sequencing data had been previously generated by TCGA[3], using either the Illumina GAIIx or HiSeq 2000 platforms. To help correct for batch effects between data platforms (GAIIx vs. HiSeq), we used the combat software[46]. Expression values within each cancer type (BLCA, KICH, KIRC, KIRP, PRAD, TGCT) were normalized to s.d. from the median across tumors. For unsupervised clustering analysis, we selected the top 500 most variable genes, according to average s.d. (using log-transformed expression values) across the six projects (for projects with both HM450 and HM27 platforms, the centroid mean of the s.d. computed for each of the two platforms was used). ConsensusClusterPlus R-package[44] was used to identify clusters in the data using 1000 iterations, 80% sample resampling from 2 to 15 clusters (k2 to k15) using hierarchical clustering with Ward as the linkage algorithm and Pearson correlation as the similarity metric. Consistent with what was carried out for the other platforms, the $k = 7$ clustering solution was selected for further investigation.

**Reverse phase protein array analysis.** RPPA data were previously generated by TCGA[3]. Raw data (level 1), SuperCurve nonparameteric model fitting on a single array (level 2), and loading corrected data (level 3) were deposited at the DCC. RPPA data were obtained from The Broad Institute's Firehose pipeline, which included data for 1570 urologic cancer cases. Expression values within each cancer type (BLCA, KICH, KIRC, KIRP, PRAD, TGCT) were normalized to s.d. from the median across tumors. For unsupervised clustering (using all 185 protein features represented in the data set), ConsensusClusterPlus R-package[44] was used to identify clusters in the data using 1000 iterations, 80% sample resampling from 2 to 15 clusters (k2 to k15) using hierarchical clustering with average linkage algorithm and Pearson correlation as the similarity metric. Consistent with what was carried out for the other platforms, the $k = 7$ clustering solution was selected for further investigation.

RPPA profiles were also scored for a previously defined[47] MAPK signature (average of phospho-SHC or pSHC, pRAF, pMEK, pERK, pSRK, pYB1, pP38, pJNK, and pJUN). For computing RPPA-based pathway score, all proteins levels were first normalized to s.d. from the median within each cancer type.

**Multiplatform-based subtype discovery.** As described above, urologic cancer cases were subtyped according to each of the individual data platforms for DNA methylation, DNA copy alteration, mRNA expression, miRNA expression, and protein expression. Subtypes defined from each platform were coded into a series of indicator variables for each subtype, with the matrix of 1 and 0 s then clustered by a Cluster of Cluster Analysis (COCA)[7, 10, 16] to define integrated subtypes. For the $k = 9$ COCA subtype solution, we defined the top differential genes associated with each subtype; we first computed the two-sided $t$-test for each gene, comparing each subtype with the rest of the tumors, then selected the top 100 genes with the lowest $P$-value for each subtype.

**Pathway and immune cell signature analysis.** Tumor expression profiles were scored for gene signatures associated with pathway deregulation essentially as previously described[10, 16], as well as outlined here. To computationally infer the infiltration level of specific immune cell types using RNA-seq data (Fig. 6b), we used a set of genes specifically overexpressed in one of 24 immune cell types from Bindea et al.[34]. Elsewhere, the Bindea signature scoring has been found to yield results consistent with those derived from immunohistochemistry (IHC) methods examining lymphocyte-specific expression patterns within cancer vs. non-cancer cellular compartments[16, 40]. For scoring TCGA cancer samples for each of these immune cell signatures, the average of the gene expression values (transformed within each cancer type to s.d. from the median) was used. In addition, samples were scored for expression of Antigen Presentation MHC class I (APM1) genes (HLA-A/B/C, B2M, TAP1/2, TAPBP) and for Antigen Presentation MHC class II (APM2) genes.

The Fantom data sets of gene expression by cell type[27] were analyzed using a previously utilized approach[16], also outlined here. Gene expression profiles from various normal human and mouse tissues[27] were obtained from the FANTOM5 data repository (http://fantom.gsc.riken.jp/5/data/); for our study, profiles from fetal or embryonic human specimens were removed from the analysis. The top 900 differential mRNAs by COCA subtype (Fig. 2b) were examined in both fantom human and fantom mouse expression data sets. Logged expression values for each gene in the fantom data set were centered on the median of sample profiles. For each fantom differential expression profile (genes centered within the fantom data set), the inter-profile correlation (Pearson's) was taken with that of each TCGA urologic cancer differential expression profile (with genes centered within each TCGA project as described above); for a given set of fantom profiles (e.g., profiles of the immune group) and a given TCGA pan-urologic genomic subtype (e.g., c1), the average correlation (represented as a $t$-statistic) between the fantom and TCGA profiles was used to represent a summary measure of overall similarity between the two groups (Fig. 6a). Fantom data set categories for immune-related or CNS-related profiles were previously defined[16]; categories for "macrophage", "mesenchymal" and "epithelial" were defined based on the name of the sample profile containing one of the above keywords.

Gene transcription signature scores associated with pathway (e.g., scores for cell cycle, p53, EMT, NRF2-ARE, hypoxia, KEGG: Glycolysis/Gluconeogenesis, KEGG: Pentose Phosphate pathway, KEGG: Fatty Acid metabolism, KEGG: TCA Cycle, and KEGG: Oxidative Phosphorylation) were computed as follows. For each gene in the TCGA pan-urologic cancer data set (combined BLCA, KICH, KIRC, KIRP, PRAD, TGCT), expression values within each cancer type were centered on the median of tumor sample profiles and divided by the s.d. across tumors. For cell cycle, p53, EMT, NRF2-ARE, hypoxia, and KEGG signatures, the average expression of the set of genes within a given signature were computed. For cyclin D1, MAP Kinase, and Yap1 signatures, urologic cancer expression profiles (with values centered to s.d. within each cancer type) were scored for the above signatures using our previously described "t-score" metric[3] (within each profile, the $t$-score was the two-sided $t$-statistic between the average of the normalized values for the "up" genes vs. the average of the normalized values for the "down" genes). Cell cycle genes were from Whitfield et al.[48]. Gene targets of p53 were from ref[49]. Gene transcription signature scores of NRF2-ARE pathway where generated as described[22], on the basis of four different signatures: "Malhotra" signatures[50], which defined expression profiling and Chip-seq of mouse embryonic fibroblasts (MEFs) with either constitutive nuclear accumulation (Keap1$^{-/-}$) or depletion (Nrf2$^{-/-}$) of Nrf2, including genes downregulated in Nrf2$^{-/-}$ vs. wild-type and Nrf2 bound, and genes upregulated in Keap1$^{-/-}$ vs. wild-type and Nrf2 bound; "GSE28230," from Gene Expression Omnibus data set of A549 adenocarcinoma lung cancer cells with siRNA knockdown of NRF2 (using $P < 0.01$, fold > 1.5); "Osburn," from GSE11287 data set of mouse liver with or without Keap1 knockout used ($P < 0.01$, fold >1.5); across the urologic cancer profiles, we normalized the individual gene signature scores to s.d. from the median across samples, and a "summary score" for NRF2-ARE pathway was computed as the average of the four individual normalized signature scores. EMT signature scores were computed as previously described[20, 51] (sum of the normalized values for ZEB1, CDH2, FN1, FOXC2, GSC, ITGB6, MMP2, MMP3, MMP9, SNAI1, SNAI2, SOX10, TWIST1, and VIM, minus the sum of the normalized values for CDH1, DSP, OCLN). Hypoxia signature was based on the set of canonical HIF1A targets from Harris[19]. Cyclin D1 and MAPK signatures were from Creighton[21]. Gene transcription signature scores of Yap1 pathway where generated as described here, on the basis of three different signatures: from GSE32567, genes differentially expressed ($P < 0.01$, fold > 1.4) with knockdown of Yap1 in hepatocellular carcinoma cell line (SK-Hep1); from GSE49406, genes differentially expressed ($P < 0.01$, fold > 1.4) with knockdown of Yap1 in HEK293 cells; from GSE7700, genes differentially expressed ($P < 0.01$, fold > 1.4) with knockdown of Yap1 in normal breast luminal cell; across the urologic cancer profiles, we normalized the individual Yap1 gene signature scores to s.d. from the median across samples, and a "summary score" for Yap1 targets was computed as the average of the three individual normalized signature scores.

**Analysis of external multi-cancer data set.** We examined an external gene expression profiling data set of multiple cancer types from the Expression Project for Oncology (expO) (GSE2109), classifying each external tumor profile by genomic subtype as defined by TCGA data. Within each cancer type, genes in the expO data set were normalized to s.d. from the median. As a classifier, the top set of 900 mRNAs distinguishing between the nine COCA subtypes in TCGA (from Fig. 2b and Supplementary Data 2) was used. For each COCA subtype the average value for each gene was computed, based on the centered TCGA expression data matrix (represented in Fig. 2b). The correlation between each expO profile and each TCGA subtype averaged profile was computed. Each of the expO bladder cases were assigned to TCGA pan-urologic subtypes c1, c2, or c8 (which were the subtypes with high representation of TCGA BLCA cases), based on which subtype profile showed the highest correlation with the given expo profile. In a similar manner, the expO prostate cases, clear cell renal cases, and papillary renal cases were also assigned to TCGA pan-urologic subtypes

**Statistical analysis.** Statistical methods regarding specific computational approaches are described above or noted in the Results. All $P$-values reported were two-sided unless otherwise noted.

**Data availability.** All data used in this study are publicly available. TCGA data are available through the Genome Data Commons (https://gdc.cancer.gov/) and the Broad Institute's Firehose data portal (https://gdac.broadinstitute.org). Somatic mutation calls made from TCGA whole exome sequencing data are available from synapse (https://www.synapse.org/#!Synapse:syn7214402). Additional molecular profiling data sets utilized as described above are available via the Gene Expression Omnibus.

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

## Acknowledgements

This work was supported in part by the National Institutes of Health (NIH) grant CA125123 (C.J.C.), and Cancer Prevention and Research Institute of Texas (CPRIT) grant RP120713 C2 (C.J.C.).

## Author contributions

Conceptualization: C.J.C.; Methodology: C.J.C., F.C., Y.Z.; Investigation: F.C., Y.Z., C.J.C., D.B., A.-K.A.L., A.A.H., J.J.H., T.K.C., D.L.G., M.I.; Formal Analysis: F.C., Y.Z., C.J.C.; Visualization; C.J.C.; Writing – Original Draft: C.J.C., Writing – Review & Editing: C.J.C., F.C., Y.Z., D.B., A.-K.A.L., A.A.H., J.J.H., T.K.C., D.L.G., M.I.; Supervision: C.J.C.

## Additional information

**Competing interests:** The authors declare no competing financial interests.

