## [Peer Review File · Nature Communications]

Reviewers' Comments:

Reviewer #1:

Remarks to the Author:

Chen et al., present NGS/proteomic data from 1954 urologic cancers representing several distinct cancer types to identify 9 major genomic subtypes using COCA analysis. By removing tissue dominant effects (mainly from RNAseq, not sure how successful this was in terms of DNA methylation, as they mention copy number effects are characteristic of tumor type and cannot be normalized) they identified common processes associated with microenvironmental influences, but not in somatic mutation patterns. They also noted differences in manifestation of specific pathways among the genomic subtypes. Overall it's a laudable effort to assemble and analyze the complexities and commonalities of the sample cohort presented in this study.

However, the study suffers from the shortcoming frequently observed in genomic analysis of high-throughput data, where it fails to narrow down the precise key features that are associated with genomic subtypes which if presented clearly could invigorate the field by fostering several follow up studies. As of now these features remain hidden in pathway patterns or in clusters that are not (for lack of a better phrase) "human readable". Accordingly it will help the studies if top candidates (based on some ranking) from the various platforms that define these genomic subtypes are clearly named and in addition provide at least some validation data as a proof of concept.

Some of the other points which need to be addressed are listed below-

1. Figure 2d numbers displayed as n (below the box plot) far exceeds the actual number of cases studied, can the authors explain this? Example BLCA adds upto 500 while the total number of cases stated in "line 99" is 412, likewise for KIRP total cases in the study is 291 but the n under the table adds upto 356 etc

2. Figure 2a, please explain how the heat map row was chosen to be indicated, does the methylation cluster in c9 of PRAD represent global hypomethylation (as indicated in fig 2d) as opposed to hyper methylation in c4? while KIRC hypermethylation is represented in heat map, the logic on how the clusters were chosen for representation is not clear, may be its just a concept and doesn't not indicate directionality. Figure 2a and more clearly so in 2b and 2c the relationship/association seemed to be forced by the clustering analysis rather than "the unifying differential patterns that span the cancers" as suggested in the text line 146. It is clearly visible both in expression and methylation (even though they have been cancer type normalized) TCGTs are in a group of their own (as expected) likewise KICH. The drastically different copy number events that characterize these cancers may be a factor that will preclude them from these types of analyses.

2. In Figure 3, survival difference of the genomic subtypes seems to mainly associate with hypermethylation status, as this association is previously shown by this group. Can this survival difference be recapitulated if analysis is performed only with the DNA methylation data? Is there an increase in specificity or sensitivity with COCA? If no, then what is the additional benefit of COCA based genomic classification?

4. In Figure 4, the somatic mutational burden of across the urologic cancers varies drastically as BLCA is known to have a heavy burden compared PRAD and RCC that are least mutated. From figure 4a it is evident that BLCA samples are mostly located towards the left where members from all pathways seem to be affected in any given sample. How does this and other dominant effects in datasets such as copy number (in KICH, TGCT) and methylation in general affect the COCA classification?

Can the authors provide some representation such as histogram perhaps, of the mutation load in the samples on top (in fig 4a)? These histograms can also be placed on subpanels in 4b to give a general sense of the mutation load context for the mutual exclusivity pattern that is emphasized by the heat map.

5. Most associations presented in figure 5 have been previously described for individual cancer type, but the evidence presented here in the heatmap is not quiet convincing that these features are common to genomic subtypes and not specific to a given cancer subtype.

6. Majority of the new subtypes are already present in previous classifications/published interrogations. For example, the authors quote that "5, c6, and c7 168 subtypes, which largely corresponded respectively to the "CC-e.3," "CC-e.2," and "CC-e.1 genomic subtypes previously associated with KIRC". I am not sure how much benefit this new classification will make or conceptually introduce.

Reviewer #2:

Remarks to the Author:

Pan-urologic cancer genomic subtypes that transcend tissue of origin

In this study the authors use multiple omics technologies (exome-seq, SNParray, RNAseq, miRNAseq, RPPA) to classify urologic cancers into nine genomic subtypes. These subtypes are thought to reflect tumor lineage-independent processes and patterns and can differ in e.g. patient survival or immune response. Subtypes can include cancer cases from different tissues (BLCA, KICH, KIRP, PRED, TGCT), indicating that subtypes of different cancer types might respond to similar treatments.

The paper is conceptually and in the used methods very similar to previous papers of the authors (which itself are based on two publications of the TCGA consortium published in 2014) that use multi-omics analysis to study subtypes of renal cell carcinoma and non-small cell lung cancer. I therefore have no general concern about the applied method and the results, but some questions regarding specific details of the chosen parameters and the follow-up analyses:

1) The process of selecting the optimal number of subtypes is not fully clear to me. The authors use each platform to generate 7 subtypes. Why 7? Does the optimal COCA cluster number change, if e.g. 3, 4, 5 or 6 subtypes are used in the platform-wise analysis?

2) In addition, the selection of 9 as optimal cluster number in COCA is not well explained. The supplement and figure S1 indicate that the area under the curve is actually better for 10 clusters. More clusters have not been tested (only 2-10). For k=10 a new cluster having only TGCT cases (formerly part of c4) is formed. Why is that solution discarded?

3) The section about differential immune profiles (lines 236 to 253) is very confusing. The top 900 differentially expressed genes (why 900?) are compared between cancer subtypes and healthy tissue data (Fantom). The authors state that this analysis indicates presence of immune response-related cells and macrophages in some subtypes, but I wonder how this enrichment was computed? Which genes are considered to define a macrophage and immune-response cell signature? Are these the 501 genes from the Bindea et al. paper that is cited 5 lines later? Based on the Supplementary Methods I think this is the case, but I cannot find out if the observed difference between subtypes is significant.

4) To continue my issues discussed in point 3), in the same paragraph the expression of immune checkpoint pathways and a list "potential targets for immunotherapy" in the subtypes is discussed. However there is no citation giving any hint what classifies these genes as targets for immunotherapy, how complete this list is, and how/why the genes were selected. The result is described as 'c5, c8, and c9 subtypes all had relatively higher expression of genes representing potential targets for immunotherapy'. So I wonder if 'all had relatively higher expression' means

'significant', or simply means 'a little bit of an increase, which is not significant'.

5) Minor: there are quite a few typos, e.g. line 149 "genes being highly expression"

In summary, apart from the section on differential immune profiles, the paper is well written and the findings are interesting (and might have impact on treatment selection in urological cancers). However the aforementioned section on immune profiles needs a rework, including better description of the used methods and analysed genes as well as a statement regarding the statistical significance of the findings.

Reviewer #1 (Expert in urological cancer) comments:

Chen et al., present NGS/proteomic data from 1954 urologic cancers representing several distinct cancer types to identify 9 major genomic subtypes using COCA analysis. By removing tissue dominant effects (mainly from RNAseq, not sure how successful this was in terms of DNA methylation, as they mention copy number effects are characteristic of tumor type and cannot be normalized) they identified common processes associated with microenvironmental influences, but not in somatic mutation patterns. They also noted differences in manifestation of specific pathways among the genomic subtypes. Overall it's a laudable effort to assemble and analyze the complexities and commonalities of the sample cohort presented in this study.

We thank the reviewer for evaluating our work.

However, the study suffers from the shortcoming frequently observed in genomic analysis of high-throughput data, where it fails to narrow down the precise key features that are associated with genomic subtypes which if presented clearly could invigorate the field by fostering several follow up studies. As of now these features remain hidden in pathway patterns or in clusters that are not (for lack of a better phrase) "human readable".

With the aim of delving further into the biology of urologic cancer subtypes, our study was carried out along the lines of that of many of the TCGA-led marker papers, where our group had contributed extensively to these papers and was well familiar with their methods and approaches. This line of investigation led us to define the processes and pathways that would distinguish the various disease subsets from each other. Commonalities observed within a number of our pan-urologic subtypes would appear to involve entire processes, versus somatic alteration of a common set of genes. In our study, we did examine individual genes for differences between the subtypes (e.g. Figures S2 and S4), and the underlying gene-level data are available as supplemental for other groups to utilize in follow-up studies (including copy alteration and mutation of specific genes being provided in Table S1, and the top 900 differential genes being provided in Table S2).

Accordingly it will help the studies if top candidates (based on some ranking) from the various platforms that define these genomic subtypes are clearly named and in addition provide at least some validation data as a proof of concept.

Figure S2 and Table S2 provide the top ranked protein, microRNA, and mRNA features. In terms of a validation, we are somewhat limited in that there would be no other large, unified multi-platform level genomic datasets equivalent to TCGA. However, we can use our top set of 900 mRNAs from Figure 2b/Table S2 to classify mRNA profiles in an external dataset, similar to what we carried out in our previous TCGA pan-Lung study (Oncogene 2016, using the external PROSPECT lung cancer mRNA dataset).

The manuscript revision includes the addition of a new main Figure (Figure 7), whereby we evaluate key patterns associated with our pan-urologic genomic subtypes in an independent, external multi-cancer expression profiling dataset, as a type of "validation." We used the Expression Project for Oncology (expO) dataset (Gene Expression Omnibus accession number GSE2109), which included mRNA expression profiles of bladder cancer (n=31 cases), prostate cancer (n=83), clear cell renal cell carcinoma (n=205), and papillary renal cell carcinoma (n=22). (There were insufficient testes and chromophobe renal cases in expO to include in our study.) To our knowledge, the expO dataset would be the largest mRNA dataset featuring multiple urologic cancer cases profiled on a common platform. While there would be fewer sample numbers in expO, and while the expO profiles are array-based rather than

RNA-seq-based, these data can well serve our purposes in determining whether the TCGA-specific patterns may be observable in an external dataset.

Figure 7 is included in this rebuttal letter below. Within each cancer type, genes in the expO dataset were first normalized to standard deviations from the median, and then each expO tumor profile was classified by pan-urolologic genomic subtype as defined by TCGA data (Figure 7a), where the top set of 900 mRNAs distinguishing between our nine COCA subtypes (Figure 2b) was used as the classifier (see Supplementary Methods). In the same manner as carried out for TCGA datasets (Figures 5 and 6), expO expression profiles were also scored for mRNA signatures related to immune cells, metabolism, hypoxia, EMT, NRF2/KEAP1, and Yap1 (Figure 7b), where similar overall trends of pathway-level differences between subtypes as originally observed in TCGA cohort could also be observed in the expO cohort.

Overall, the above represents a nice result for us to add to the current study. We appreciate the Reviewer's suggestion.

Some of the other points which need to be addressed are listed below-

1. Figure 2d numbers displayed as n (below the box plot) far exceeds the actual number of cases studied, can the authors explain this? Example BLCA adds upto 500 while the total number of cases stated in "line 99" is 412, likewise for KIRP total cases in the study is 291 but the n under the table adds upto 356 etc

We thank the reviewer for catching this! The plot itself is correct, but there were typos involving three numbers. These have been corrected in the revision. There are 1861 cases in all represented by Figure 2d.

2. Figure 2a, please explain how the heat map row was chosen to be indicated, does the methylation cluster in c9 of PRAD represent global hypomethylation (as indicated in fig 2d) as opposed to hyper

methylation in c4? while KIRC hypermethylation is represented in heat map, the logic on how the clusters were chosen for representation is not clear, may be its just a concept and doesn't not indicate directionality.

The reviewer is correct: Figure 2a does not indicate directionality of change (e.g. hypo- versus hyper-methylation, which information is provided by the other Figure components). As noted in Figure 2a legend: "The heat map displays the subtypes defined independently by DNA methylation (pink), Chromosomal copy alteration (black), mRNA expression (red), microRNA expression (blue), and protein (RPPA) expression (green); each row in this heat map denotes membership within a specific subtype defined by the indicated platform."

Figure 2a follows the same presentation template as that of our previous studies. Our present study was carried out along the lines of that of many of the TCGA-led marker papers, as well as that of our own recent papers carrying our multiplatform analyses of renal and lung cancers, respectively (Cell Reports 2016, Oncogene 2016). First, we carry our unsupervised clustering to define the nine genomic subtypes. Then, we identify the top differential features and pathways that would characterize each subtype, which can provide insight into the biology underlying a given disease subset.

Figure 2a and more clearly so in 2b and 2c the relationship/association seemed to be forced by the clustering analysis rather than "the unifying differential patterns that span the cancers" as suggested in the text line 146.

By design, the top differential features for gene expression and for methylation, according to genomic subtype (as defined using the COCA matrix presented in Figure 2a) are defined in Figures 2b and 2c, respectively. These top differential features are, in fact, defined using the clustering analysis results. More differential features that would also be statistically significant could be defined as well using the underlying data, even if we limit ourselves here to the top 100 mRNAs for each subtype in Figure 2b, for example.

It is clearly visible both in expression and methylation (even though they have been cancer type normalized) TCGTs are in a group of their own (as expected) likewise KICH.

TGCT.c4 tumors do share common expression patterns with those of other tumors in the c4 group, as do TGCT.c6 tumors with those of other tumors in the c6 group (Figure 2b). Similarly, KICH share some common patterns with those of other c1 tumors. Within a given pan-urolologic subtype, cancers of a given type may also show some distinctive patterns from those of the other tumors.

The drastically different copy number events that characterize these cancers may be a factor that will preclude them from these types of analyses.

We would consider this observation a finding, as noted in the results: "DNA copy alteration subtypes were distinguishable on the basis of cancer type rather than genomic subtype (Figure 2a)." Copy number alteration patterns are lineage-specific in our analysis results. As just one of the five platforms included as part of the standard COCA subtyping, copy number would not be having an undue influence on the unsupervised clustering results.

3. In Figure 3, survival difference of the genomic subtypes seems to mainly associate with hypermethylation status, as this association is previously shown by this group. Can this survival difference be recapitulated if analysis is performed only with the DNA methylation data? Is there an increase in specificity or sensitivity with COCA? If no, then what is the additional benefit of COCA based genomic classification?

In the revision, we have included a new figure (Figure S3), which shows the differences in patient overall survival, according to the presence or absence of high overall DNA methylation patterns (Figure also shown below).

In line with observations made in our previous study (Chen, Zhang, Cell Reports 2016), widespread DNA hypermethylation patterns are associated here with poor patient outcome for both KIRC and KIRP cohorts, but with no similar association being observed for BLCA cohort (Figure S3, there being insufficient follow-up data for evaluating the other cancer types).

The overall goal of the COCA subtyping would not be that of defining the “best” prognostic classifier. This would represent another avenue of investigation and other analytical approaches. The overall survival differences would seem to reflect inherent differences in the biology of the respective subtypes. Importantly, our COCA-based molecular subtypes are found to show excellent concordance with molecular subtyping results of the previous TCGA studies marker studies focusing on individual cancer types. Given the various analytical methods and data platforms that might be used to classify cancers, the strong overlaps between previous classifications and those made by our study would serve to better define the precise number of biologically distinct, global molecular subtypes that may be found within each urologic cancer type.

4. In Figure 4, the somatic mutational burden of across the urologic cancers varies drastically as BLCA is known to have a heavy burden compared PRAD and RCC that are least mutated. From figure 4a it is evident that BLCA samples are mostly located towards the left where members from all pathways seem to be affected in any given sample. How does this and other dominant effects in datasets such as copy number (in KICH, TGCT) and methylation in general affect the COCA classification?

Mutation was not one of the platforms used in the COCA classification, so would have no direct bearing on the classification results. DNA mutation events are typically too sparse for these to be effective in classifying tumor samples. Copy number alteration patterns are lineage-specific in our analysis results, as noted in the results. As just one of the five platforms included as part of the standard COCA subtyping, copy number would not be having an undue influence on the unsupervised clustering results.

In addition, it should be noted that our COCA-based genomic subtypes are found to show excellent concordance with results of previous studies (Figure 3a) and would show evidence of distinctive biology on the basis of pathway and integrative analyses. Given the various analytical methods and data platforms that might be used to classify cancers, the strong overlaps between previous classifications and those made by our study would serve to better define the precise number of biologically distinct, global molecular subtypes that may be found within each cancer type.

Can the authors provide some representation such as histogram perhaps, of the mutation load in the samples on top (in fig 4a)? These histograms can also be placed on subpanels in 4b to give a general sense of the mutation load context for the mutual exclusivity pattern that is emphasized by the heat map.

In the revised Figures 4a and 4b, we have added data tracks for the number of mutations per megabase for each sample represented. As expected, BLCA cases would tend to have the highest overall numbers of mutations.

5. Most associations presented in figure 5 have been previously described for individual cancer type, but the evidence presented here in the heatmap is not quiet convincing that these features are common to genomic subtypes and not specific to a given cancer subtype.

To our eyes, the heat map patterns representing key features for a given subtype (e.g. the features also noted in main Table 1) look sufficiently uniform across samples within that particular subtype. There is variation across all the samples, as expected. Not all of the features shown would necessarily need to be uniform for all subtypes. Protein features may also vary, but downstream transcriptional patterns may appear more uniform. Also, for HIPPO pathway, different upstream regulators (e.g. LATS, NF2, SAV1) may be expressed differently in different samples, with YAP1 transcription targets representing a common downstream event.

In addition to the heat map representation of Figure 5, Supplemental Figure S5 (parts a and b also shown below) shows boxplot representations of the key features, where we see that for key comparisons as indicated, the patterns appear common to multiple cancer types existing within the given subtype.

6. Majority of the new subtypes are already present in previous classifications/published interrogations. For example, the authors quote that “5, c6, and c7 168 subtypes, which largely corresponded respectively to the “CC-e.3,” “CC-e.2,” and “CC-e.1 genomic subtypes previously associated with KIRC”. I am not sure how much benefit this new classification will make or conceptually introduce.

Our results provide a framework by which we can relate disease subsets of different urologic cancer types to each other. This would offer opportunities to apply what has been previously learned about one subtype to another subtype, where both subtypes may originate from different tissues or cells of origin but yet associate with each other as part of a pan-urologic subtype. By this approach, for example, the aggressive subset of kidney papillary cancer was found here to have similar pathway-level associations as those of aggressive kidney clear cell cancer. Overall, the commonalities observed within a number of

our pan-urogenic subtypes would appear to have more to do with microenvironmental influences (e.g. cancer-associated macrophages and lymphocytes, altered metabolism, hypoxia, cellular stress), than with somatic mutation patterns, which is a key finding.

In the clinical setting, it is unlikely that patients would be evaluated for five different genomic data platforms in the same way as was done with the TCGA cases. However, individual molecular markers of our urogenic cancer subtypes that might seem most relevant from a therapeutic standpoint could potentially be evaluated in the setting of patient care in future studies.

Reviewer #2 (Expert in bioinformatics) comments:

In this study the authors use multiple omics technologies (exome-seq, SNParray, RNAseq, miRNAseq, RPPA) to classify urogenic cancers into nine genomic subtypes. These subtypes are thought to reflect tumor lineage-independent processes and patterns and can differ in e.g. patient survival or immune response. Subtypes can include cancer cases from different tissues (BLCA, KICH, KIRP, PRED, TGCT), indicating that subtypes of different cancer types might respond to similar treatments.

The paper is conceptually and in the used methods very similar to previous papers of the authors (which itself are based on two publications of the TCGA consortium published in 2014) that use multi-omics analysis to study subtypes of renal cell carcinoma and non-small cell lung cancer. I therefore have no general concern about the applied method and the results, but some questions regarding specific details of the chosen parameters and the follow-up analyses:

We thank the reviewer for evaluating our work.

1) The process of selecting the optimal number of subtypes is not fully clear to me. The authors use each platform to generate 7 subtypes. Why 7? Does the optimal COCA cluster number change, if e.g. 3, 4, 5 or 6 subtypes are used in the platform-wise analysis?

Using the $k=7$ solution for each data platform was arrived at based on objective assessments of cluster stability, along with the consideration of results from prior studies. In the revised manuscript, we have added an additional figure (Figure S1a, also shown below), which shows, for each data platform, the Delta area plot graphic showing the relative change in area under the CDF curve comparing k and $k - 1$. This graphic allows one to determine the relative increase in consensus and determine k at which there is no appreciable increase.

For consistency, we wanted to choose the same k uniformly for all platforms, and after $k=7$ there is no appreciable increase in consensus for any given platform. Solutions of 3, 4, or 5 may represent too few platform-level subtypes according to the above. A solution of $k=6$ likely would yield a solution similar to what we present in the paper, as the COCA step provides another opportunity for merging and consensus.

Every genomic subtyping study would utilize its own analytic approach and decision points; while here we found the final solution presented to represent a good framework for the exploration of differences

between the urologic cancer subtypes. Importantly our genomic molecular subtypes are found to show excellent concordance with results of previous studies utilizing other analytical approaches and platforms (Figure 3a), and so our COCA-based subtypes would not be expected to be highly sensitive to minor tunings of the parameters used.

2) In addition, the selection of 9 as optimal cluster number in COCA is not well explained. The supplement and figure S1 indicate that the area under the curve is actually better for 10 clusters. More clusters have not been tested (only 2-10). For $k=10$ a new cluster having only TGCT cases (formerly part of c4) is formed. Why is that solution discarded?

Every genomic subtyping study would utilize its own analytic approach and decision points; in our case, the choice of using $k=9$ for our COCA solution was deliberate. At $k=9$, BLCA cases are subdivided between three subtypes, and a fourth KIRP subtype (c3) is defined. At $k=10$, the $k=9$ c4 TGCT cases form their own subtype distinct from that of the other c4 cases (Figure S1e). At higher k (e.g. $k \geq 10$), the individual cancer types within a given COCA $k=9$ subtype will then separate from the other cases (just as we observed for the TGCT.c4 cases as $k=10$), due in part to the copy number clustering (which is more cancer type-specific).

We wanted to use a solution where the subtypes featured cancers of different types. This would allow for us to identify the common processes driving cellular behavior across tumor lineages, which was a stated goal of the study. TGCT.c4 tumors do share common expression patterns with those of other tumors in the c4 group, even though within any given pan-urologic subtype, cancers of a given type may also show some distinctive patterns from those of the other tumors. With higher k , the CDF will continue to reach towards an approximate maximum, but after a point there are diminishing returns with using a larger number of subtypes, which makes generalizations more difficult. While additional molecular subtypes could be defined arbitrarily (e.g. by increasing the COCA k), these would also need to have some unique biology associations for them to be relevant.

3) The section about differential immune profiles (lines 236 to 253) is very confusing. The top 900 differentially expressed genes (why 900?) are compared between cancer subtypes and healthy tissue data (Fantom). The authors state that this analysis indicates presence of immune response-related cells and macrophages in some subtypes, but I wonder how this enrichment was computed?

We have expanded the Methods section in the main manuscript, to better describe the analysis involving the fantom dataset: "Briefly, the top 900 differential mRNAs by COCA subtype (Figure 2b) were examined in both fantom human and fantom mouse expression datasets. Logged expression values for each gene in the fantom dataset were centered on the median of sample profiles. For each fantom differential expression profile (genes centered within the fantom dataset), the inter-profile correlation (Pearson's) was taken with that of each TCGA urologic cancer differential expression profile (with genes centered within each TCGA project as described above); for a given set of fantom profiles (e.g. profiles of the immune group) and a given TCGA pan-urologic genomic subtype (e.g. c1), the average correlation (represented as a t -statistic) between the fantom and TCGA profiles was used to represent a summary measure of overall similarity between the two groups (Figure 6a)." In summary, we used the same approach to the fantom dataset, as what was used in our recent TCGA pan-lung paper (Oncogene 2016). The 900 genes were selected based on the set from Figure 2b, which genes best showed the differences between the pan-urologic cancer subtypes (though using another set of highly variable genes or even all the genes in the dataset should yield similar overall results, based on our experience). As noted in the supplementary methods, Fantom dataset categories for immune-related or CNS-related profiles were previously defined in our TCGA pan-lung paper; categories for "macrophage", "mesenchymal" and "epithelial" were defined based on the name of the sample profile containing one of the above keywords.

Which genes are considered to define a macrophage and immune-response cell signature? Are these the 501 genes from the Bindea et al. paper that is cited 5 lines later? Based on the Supplementary Methods I think this is the case, but I cannot find out if the observed difference between subtypes is significant.

Two different approaches were used in assessing the levels of macrophage- and immune cell-related signatures in TCGA tumors. In the revision of the main text, we have edited the results and methods to better distinguish between the two approaches—fantom vs Bindea—used here. Regarding the fantom correlations, in the revised manuscript Figure legend for 6a, we now note the following: “For “immune,” “macrophage”, and “mesenchymal” correlations, differences are significant with $p < 1E-10$ (t-test), when comparing c5 with c6 cases, comparing c8 with c1 cases, and comparing c9 with c2 cases.”

Regarding the Bindea et al. signatures, we have added another supplementary figure (Figure S6, also shown below, highlighting the Bindea immune-related expression signatures across pan-urolologic genomic subtypes, where in this case the individual genes constituting each signature are represented. (Of the entire set of cell-specific genes from Bindea et al., 485 were represented in TCGA RNA-seq datasets as indicated in the figure.) For each tumor profile, the normalized values within a given signature were averaged in order to derive a signature score (as shown in Figure 6b).

4) To continue my issues discussed in point 3), in the same paragraph the expression of immune checkpoint pathways and a list “potential targets for immunotherapy” in the subtypes is discussed. However there is no citation giving any hint what classifies these genes as targets for immunotherapy, how complete this list is, and how/why the genes were selected.

In the revision, we have added a reference in the results to our pan-RCC paper (Cell Reports 2016), from which we first presented the list of potential targets for immunotherapy. This list of genes was originally

provided to us by the study's co-authors who also happened to be experts in this field. Figure 6d also provides information on additional genes related to immune checkpoint pathway.

The result is described as 'c5, c8, and c9 subtypes all had relatively higher expression of genes representing potential targets for immunotherapy'. So I wonder if 'all had relatively higher expression' means 'significant', or simply means 'a little bit of an increase, which is not significant'.

We have revised Figure 6b to indicate which features were significant (with $p < 0.01$) for comparisons of interest (i.e. the comparisons highlights in Figures 6c and 6d). All features shown were significant for at least one comparison.

5) Minor: there are quite a few typos, e.g. line 149 "genes being highly expression"

We have checked throughout the revised manuscript and corrected typos where found.

In summary, apart from the section on differential immune profiles, the paper is well written and the findings are interesting (and might have impact on treatment selection in urological cancers). However the aforementioned section on immune profiles needs a rework, including better description of the used methods and analysed genes as well as a statement regarding the statistical significance of the findings.

We greatly appreciate the positive assessment regarding our study. We hope that the revised methods and results write-up involving the immune signature analysis (as described above regarding Reviewer comment #3), and the addition of statistics to Figures 6a and 6b (in response to comment #4) will satisfy any outstanding concerns.

Reviewers' Comments:

Reviewer #2:

Remarks to the Author:

In the previous review my main concerns were a) the choice of k for the number of subtypes and b) the lack of detailed method descriptions and significance levels for the immune cell signature analysis. Both issues have been addressed by the authors.

Regarding the choice of subtypes the author's arguments are convincing and the new supplementary figures showing CDF for single platforms is helpful.

Regarding the analysis of immune cell signatures the authors have substantially added to the methods description and provided p-values.

My only remaining criticism is that the definition of "potential targets for immunotherapy" is still not clear. The list was defined for the previous paper (Chen Cell Report 2016), but in that paper there is also no indication of how these genes have been selected. (Similar to this paper there is only a single sentence.) Obviously the previous reviewers had no issue with that choice. But simply stating that 'This list of genes was originally provided to us by the study's co-authors who also happened to be experts in this field' is quite a weak argument. I would suggest to cite some review paper of these (or other) experts.

Reviewer #3:

Remarks to the Author:

The authors performed additional analyses to answer the reviewers comments. The response for the previous review is very detailed and acceptable.

Reviewer #2 comments:

My only remaining criticism is that the definition of “potential targets for immunotherapy” is still not clear. The list was defined for the previous paper (Chen Cell Report 2016), but in that paper there is also no indication of how these genes have been selected. (Similar to this paper there is only a single sentence.) Obviously the previous reviewers had no issue with that choice. But simply stating that 'This list of genes was originally provided to us by the study's co-authors who also happened to be experts in this field' is quite a weak argument. I would suggest to cite some review paper of these (or other) experts.

We thank the reviewer for evaluating our work. In the revision, we have added the following references to the Results section involving the genes examined in Figure 6B, which collectively provide strong support for each of these genes as representing potential targets for immunotherapy:

Chen, D. & Mellman, I. Elements of cancer immunity and the cancer-immune set point. Nature 541, 321-330 (2017).

Hsieh, J., et al. Renal cell carcinoma. Nat Rev Dis Primers 3, 17009 (2017).

Linch, S., McNamara, M. & Redmond, W. OX40 Agonists and Combination Immunotherapy: Putting the Pedal to the Metal. Front Oncol 5, 34 (2015).

LaRue, H., Ayari, C., Bergeron, A. & Fradet, Y. Toll-like receptors in urothelial cells--targets for cancer immunotherapy. Nat Rev Urol 10, 537-545 (2013).

Salmaninejad, A., et al. Cancer/Testis Antigens: Expression, Regulation, Tumor Invasion, and Use in Immunotherapy of Cancers. Immunol Invest 45, 619-640 (2016).

Lou, Y., et al. Epithelial-Mesenchymal Transition Is Associated with a Distinct Tumor Microenvironment Including Elevation of Inflammatory Signals and Multiple Immune Checkpoints in Lung Adenocarcinoma. Clin Cancer Res 22, 3630-3642 (2016).